# LoRA-Ensemble: Efficient Uncertainty Modelling for Self-attention Networks

## Abstract

Numerous crucial tasks in real-world decision-making rely on machine learning algorithms with calibrated uncertainty estimates. However, modern methods often yield overconfident and uncalibrated predictions. Various approaches involve training an ensemble of separate models to quantify the uncertainty related to the model itself, known as epistemic uncertainty. In an explicit implementation, the ensemble approach has high computational cost and high memory requirements. This particular challenge is evident in state-of-the-art neural networks such as transformers, where even a single network is already demanding in terms of compute and memory. Consequently, efforts are made to emulate the ensemble model without actually instantiating separate ensemble members, referred to as implicit ensembling. We introduce LoRA-Ensemble, a parameter-efficient deep ensemble method for self-attention networks, which is based on Low-Rank Adaptation (LoRA). Initially developed for efficient LLM fine-tuning, we extend LoRA to an implicit ensembling approach. By employing a single pre-trained self-attention network with weights shared across all members, we train member-specific low-rank matrices for the attention projections. Our method exhibits superior calibration compared to explicit ensembles and achieves similar or better accuracy across various prediction tasks and datasets.

## 1 Introduction

Machine learning models are increasingly applied also in fields where incorrect estimates may have severe consequences, e.g., autonomous driving, medical diagnosis, (extreme) weather event prediction, or agricultural management decision support. In such applications well-calibrated predictive uncertainties are crucial to enable self-diagnosis. Uncertainty can be separated into two components. *Aleatoric uncertainty*, a.k.a. irreducible noise, is inherent in the data. *Epistemic uncertainty* on the other hand stems from a lack of knowledge about certain regions of the input space, due to a lack of training data (Der Kiureghian & Ditlevsen, 2009).

Quantification of epistemic uncertainty in large machine learning models is non-trivial. Analytical computation is usually intractable, thus research has focused on efficient approximations (Graves, 2011; Blundell et al., 2015; Welling et al., 2011). To date, probabilistic ensembles remain the best-performing approach (Lakshminarayanan et al., 2017). In a naïve implementation, such an ensemble consists of multiple independently trained models. Individual models are interpreted as Monte Carlo samples from the posterior weight space and are used to obtain an unbiased estimator of the posterior distribution. To achieve a low correlation between ensemble members one can capitalize on the stochastic nature of the training process and start from different initial weights, and/or sample different random batches of data. The basic principle is that the predictions of different ensemble members will agree near observed training samples, whereas they may vary far away from the training data. Their spread therefore serves as a measure of epistemic uncertainty. Even small ensembles often capture the uncertainty well (in expectation), i.e., they are well calibrated.

An issue with naïve ensembles is that their computational cost and memory footprint grow proportionally to the number of ensemble members. For smaller models explicit ensembling may still be feasible, albeit with higher financial cost and energy consumption. For modern neural networks with up to several billion parameters, hardware restrictions render the naïve approach intractable, in particular, one can no longer hold the entire ensemble in memory. Consequently, a lot of research has

gone into ways of creating ensembles implicitly, without requiring multiple copies of the full base model (Wen et al., 2020; Wenzel et al., 2020; Huang et al., 2017; Turkoglu et al., 2022). Unfortunately, most of these parameter-efficient ensembling techniques are not applicable to the newest generation of neural networks. Transformer networks (Vaswani et al., 2017) have recently become popular due to their superior ability to capture complex structures in data. However, implicit ensembling schemes tend to underperform for transformers or are incompatible with them, as detailed in Appendix O.

Several studies have shown that modern neural networks are heavily overparametrized and that their results have low intrinsic dimension (Li et al., 2018a; Aghajanyan et al., 2020). This led Hu et al. (2021) to propose Low-Rank Adaptations (LoRAs) as a way of deploying individually fine-tuned Large Language Models (LLMs) to different tasks while avoiding the prohibitively large memory and compute requirements of retraining them. It turns out that the weight matrices in such models can be factorized to have very low rank, with hardly any loss in prediction performance.

This led us to use LoRA as a basis for a novel, parameter-efficient ensemble method tailored to transformer architecture. In line with the trend towards transfer learning, our method uses a pre-trained transformer model, which is expanded into an implicit ensemble by varying the LoRA factorization, while keeping backbone weights frozen. In this way, our method requires a small number of additional parameters to turn an existing transformer model into a diverse ensemble whose performance across various tasks is comparable to an Explicit Ensemble. In summary, our contributions are:

- We introduce LoRA-Ensemble, a parameter-efficient probabilistic ensemble method for self-attention networks.
- LoRA-Ensemble can be readily combined with most pre-trained transformer networks, irrespective of their specific architecture and application domain: it simply replaces the linear projection layers in the attention module with LoRA-Ensemble layers.
- We apply LoRA-Ensemble to different classification tasks including: conventional image labeling, classification of skin lesions in dermatoscopic images, sound classification from spectrograms, and out-of-distribution (OOD) detection. In these experiments, LoRA-Ensemble not only consistently outperforms other implicit ensemble schemes but also, surprisingly, its classification accuracy and uncertainty calibration are often even better than that of an Explicit Ensemble.
- We analyze the superior performance of LoRA-Ensemble compared to Explicit Ensemble by exploring the diversity of ensemble members in both function and weight spaces.

## 2 LoRA-Ensemble

The Low-Rank Adaptation (LoRA) technique makes it possible to use a pre-trained model and fine-tune it without having to retrain all its parameters. This is particularly beneficial for modern neural networks with large parameter spaces. The underlying principle is to freeze the pre-trained model weights $W_0 \in \mathbb{R}^{k \times d}$ and instead constrain the updates to a low-rank decomposition. This can be expressed mathematically as:

$$W = W_0 + \Delta W = W_0 + B \cdot A \ . \tag{1}$$

Here $B \in \mathbb{R}^{k \times r}$ and $A \in \mathbb{R}^{r \times d}$ are two trainable low-rank matrices, where $r \ll \min(d, k)$. $W$ and $\Delta W$ are then multiplied with the same input $x$, which yields the following modified forward pass:

$$h = W_0 \cdot x + \Delta W \cdot x = W_0 \cdot x + B \cdot A \cdot x \ . \tag{2}$$

LoRA applies this low-rank updating scheme only to weights in the self-attention modules of a transformer model while leaving the interleaved MLP modules untouched. I.e., the weight matrices being updated are $W_q$, $W_k$, and $W_v$ for the query, key, and value of the attention mechanism, as well as the $W_o$ for merging the multi-head outputs. The former three are each treated as a single matrix, disregarding the fact that they are typically sliced into multiple attention heads. (Hu et al., 2021)

Although not designed with uncertainty calibration in mind, the LoRA concept fulfills all the requirements of an implicit deep ensemble: By modifying the weights of the highly nonlinear self-attention mechanism one is able to generate a diverse collection of networks with the same architecture and objective. By learning an additive, low-rank update $\Delta W = B \cdot A$ rather than directly

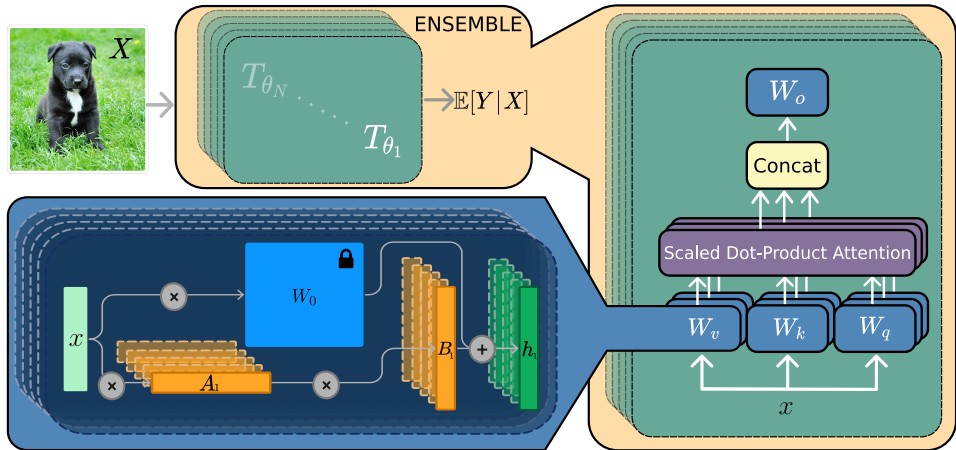

Figure 1: A schema of a LoRA-Ensemble. The computation structure of the multi-head self-attention module (right), and LoRA-Ensemble module (bottom left). $X$ denotes the actual input, and $x$ represents the intermediate input representation.

tuning the weight matrices, the expansion into a model ensemble adds only a small number of parameters and is efficient. In detail, we start from a single, pre-trained model with frozen parameters $W_0$ and expand it with a set of trainable low-rank matrices $\Delta W_i, \forall i = 1 \dots N$. At each transformer block, there now is a different forward pass per ensemble member $i$, as illustrated in Fig. 1:

$$h_i = W_0 \cdot x + \Delta W_i \cdot x = W_0 \cdot x + B_i \cdot A_i \cdot x \ , \tag{3}$$

leading to $N$ different predictions $T_{\theta_i}(X)$ for a given input $X$. From those individual predictions, we compute the ensemble estimate by simple averaging:

$$\mathbb{E}[Y|X] = \frac{1}{N} \sum_{i=1}^{N} T_{\theta_i}(X) \ . \tag{4}$$

## 2.1 IMPLEMENTATION

In practice, our LoRA-Ensemble is implemented by replacing the respective linear layers ($W_q$, $W_k$, $W_v$, and $W_o$) in the pre-trained model architecture with custom LoRA modules.

As a backbone for experiments with image datasets, we employ a Vision Transformer (ViT) model (Dosovitskiy et al., 2020). The chosen architecture is the *base* variant with patch size $32 \times 32$ as defined in Dosovitskiy et al. (2020). We load the weights from `torchvision`, which were trained on ImageNet-1k (Deng et al., 2009), using a variant of the training recipe from Touvron et al. (2020), for details refer to their documentation.

The forward pass through the backbone is parallelized by replicating the input along the batch dimension. In each LoRA module, the data is split into separate inputs per member and passed to the respective member with the help of a *vectorized map*, which allows a parallelized forward pass even through the LoRA modules. The outputs are then again stacked along the batch dimension. In this way, one makes efficient use of the parallelization on GPU, while at the same time avoiding loading the pre-trained backbone into memory multiple times.

As a backbone for audio experiments, we use the Audio Spectrogram Transformer (AST) backbone (Gong et al., 2021). That architecture was inspired by ViT (more specifically the data-efficient version of ViT akin to DeiT (Touvron et al., 2020)) but is designed specifically for audio spectrograms. Following Gong et al. (2021), we initialize the audio model weights by transferring and appropriately interpolating them from ImageNet pre-training. See Appendix I and J for details. As the AST version of LoRA-Ensemble would run into memory limits, we introduce chunking. While the forward pass through the backbone is still parallelized, the LoRA modules are called sequentially.[1]

---

[1]For the Explicit Ensemble the vectorization could not be used on GPU, due to a technical issue with the ViT implementation in PyTorch.

Finally, the pre-trained model does not have the correct output dimension for our prediction tasks (i.e., it was trained for a different number of classes). Therefore we entirely discard its last layer and add a new one with the correct dimensions, which we train from scratch. Obviously, the weights of that last layer are different for every ensemble member. We parallelize it in the same way as the LoRA module described above.

A PyTorch implementation of LoRA-Ensemble, as well as pre-trained weights to reproduce the experiments, will be publicly released on GitHub.

## 3 EXPERIMENTS

In the following section, we evaluate the proposed LoRA-Ensemble on several datasets with regard to its predictive accuracy, uncertainty calibration, and memory usage. For each experiment we also show 1-sigma error bars, estimated from five independent runs with different random initializations.

As a first sandbox experiment, we perform image classification for the popular, widely used CIFAR-100 benchmark (Krizhevsky, 2009) (see Appendix A for CIFAR-10 experiment). The dataset consists of 100 object classes, each with 600 samples, for a total size of 60 000 images. From that set, 10 000 images are designated test data, with all classes equally distributed between the training and testing portions.

The HAM10000 dataset was proposed for the *Human Against Machine with 10 000 training images* study (Tschandl et al., 2018). It consists of 10 015 dermatoscopic images of pigmented skin lesions, collected from different populations. The dataset was initially assembled to compare machine learning methods against medical professionals on the task of classifying common pigmented skin lesions. Compared to CIFAR-100, this is arguably the more relevant test bed for our method: in the medical domain, uncertainty calibration is critical, due to the potentially far-reaching consequences of incorrect diagnoses and treatment planning.

For both datasets, LoRA-Ensemble is compared against several baselines. As a sanity check, we always include results obtained with a single Vision Transformer (ViT) model, as well as with a single ViT model with LoRA in the attention modules. These models do not have a dedicated mechanism for uncertainty calibration, instead, the predicted class-conditional likelihoods are used to quantify uncertainty. Furthermore, we compare to an explicit model ensemble, Monte Carlo Dropout (MC Dropout) as implemented in Li et al. (2023) and a modified version of Snapshot Ensemble (Huang et al., 2017), detailed in Appendix N. Snapshot Ensemble is the only well established implicit ensembling technique that is architecture agnostic and can therefore be applied to self-attention networks in a straightforward fashion. For implementation challenges of other implicit methods, please refer to Appendix O. The LoRA rank was empirically set to 8 for CIFAR-100 and 4 for HAM10000.

We evaluate predictive performance and calibration quality for each method using multiple metrics. Predictive accuracy is assessed with classification accuracy (percentage of correctly classified test samples) and the F1-score, which balances precision and recall. Calibration quality is measured using the Expected Calibration Error (ECE), Negative Log-Likelihood (NLL), and Brier score. The ECE quantifies the deviation from a perfectly calibrated model, i.e., one where the estimated uncertainty of the maximum-likelihood class correctly predicts the likelihood of a miss-classification. Definitions of all metrics are provided in Appendix P.

As a further benchmark from a different application domain, we process the ESC-50 environmental sounds dataset (Piczak, 2015). It consists of 2000 sound samples, each five seconds long, that represent 50 different semantic classes with 40 samples each. To prepare the raw input waveforms for analysis, they are converted into 2-dimensional time/frequency spectrograms, see Gong et al. (2021). These spectrograms form the input for the Audio Spectrogram Transformer, a state-of-the-art transformer model for sound classification.

As for the ViT model, we train an Audio Spectrogram Transformer version of LoRA-Ensemble by modifying the attention weights with different sets of LoRA weights. That ensemble is then compared to a single instance of AST with and without LoRA, to an Explicit Ensemble of AST-models, and to an MC Dropout variant of AST, similar to Li et al. (2023). For ESC-50 a LoRA rank of 16 worked best, presumably due to the larger domain gap between (image-based) pre-training and the

Table 1: Parameter counts and computation times for an Explicit Ensemble of 16 ViT models and the corresponding LoRA-Ensemble. Training time is the average duration for one epoch on CIFAR-100, with batch size 32. Inference time is the average duration of a forward pass, with batch size 1.

| Method | Parameter overhead | Training time [s] | Inference time [ms] |
|---|---|---|---|
| Explicit Ensemble | $16 \times 87M$ | $16 \times 139$ | $16 \times 4.6$ |
| LoRA-Ensemble | $1.12 \times 87M$ | $1108$ | $22.7$ |

actual audio classification task. The experimental evaluation in Gong et al. (2021) employs the same performance metrics as before, but a slightly different evaluation protocol. Model training (and evaluation) is done in a 5-fold cross-validation setting, where the epoch with the best average accuracy across all five folds is chosen as the final model. The performance metrics given below are calculated by taking the predictions of all five folds at the chosen epoch and evaluating accuracy and calibration metrics jointly. While the accuracy calculated this way is equivalent to the average of all five folds, others are not, so this method results in a more realistic calculation of the calibration metrics.

For the out-of-distribution (OOD) detection experiment, we trained models on the CIFAR-100 dataset and evaluated their performance using samples from CIFAR-100 (in-distribution) and CIFAR-10 or SVHN Netzer et al. (2011) (out-of-distribution), following standard OOD detection practices Hendrycks & Gimpel (2016). We assessed the performance by calculating the area under the ROC curve (AUROC) and the area under the precision-recall curve (AUPRC).

## 3.1 COMPUTATIONAL COST

In addition to evaluating classification performance and calibration, we assess the computational cost in terms of parameters, training, and inference time. The required resources are presented in Tab. 1.

The total *number of parameters* is reported for an ensemble of 16 members, and matrices $A$ and $B$ with rank 8 when using LoRA. Choosing a different rank will slightly alter the parameter count. In many cases a lower rank may suffice, cf. Hu et al. (2021). All times were measured on a single NVIDIA Tesla A100-80GB GPU. *Training time* is given as the average wall clock time per training epoch on CIFAR-100, with 16 ensemble members. *Inference time* is computed as the average time for a single forward pass for a CIFAR-100 example, with batch size 1. As mentioned in Sec. 2.1, the forward pass for the Explicit Ensemble processes the members sequentially.[2] Hence, we calculate the average time needed for one member and multiply it by 16. It is evident that the proposed method uses significantly fewer parameters and less memory. LoRA-Ensemble also trains faster, and speeds up inference more than 3 times.

We point out that, with our current implementation, the runtime comparisons are still indicative. It turns out that PyTorch's vectorized map (vmap) has a large one-time overhead that is only amortized when using large ensembles, while small ensembles are slowed down. Practical ensemble sizes will benefit when implemented in a framework that supports just-in-time compilation, like JAX.

## 3.2 CIFAR-100

Quantitative results are summarized in Tab. 2. Reliability diagrams, along with plots depicting classification accuracy and ECE as a function of ensemble size, are provided in Appendix A.2.

LoRA-Ensemble consistently reaches higher accuracy than MC Dropout and Snapshot Ensemble, with a notable edge of approximately 5 percentage points for ensembles of four or more members. Surprisingly, it also consistently surpasses the Explicit Ensemble by about 2 percentage points, apparently a consequence of the fact that already a single ViT model, and thus every ensemble member, benefits from the addition of LoRA.

With LoRA-Ensemble also the estimates of predictive uncertainty are better calibrated. Interestingly, the calibration of a single network with LoRA is already very good but slightly degrades when

---

[2]Speed comparisons only make sense with the same resources. With sufficiently many GPUs any ensemble method can be parallelized by instantiating explicit copies of different members on separate GPUs.

Table 2: Model performance on the CIFAR-100 dataset for the compared methods. Ensembles have 16 members. Best score for each metric in **bold**, second-best underlined.

| Method | Accuracy ($\uparrow$) | F1 ($\uparrow$) | ECE ($\downarrow$) | NLL ($\downarrow$) | Brier ($\downarrow$) |
|---|---|---|---|---|---|
| Single Network | $76.6 \pm 0.3$ | $76.6 \pm 0.3$ | $0.145 \pm 0.004$ | $1.181 \pm 0.019$ | $0.370 \pm 0.004$ |
| Single Net w/ LoRA | $79.6 \pm 0.2$ | $79.4 \pm 0.2$ | $\mathbf{0.014} \pm 0.003$ | $\underline{0.671} \pm 0.005$ | $0.286 \pm 0.003$ |
| MC Dropout | $77.1 \pm 0.5$ | $77.2 \pm 0.4$ | $0.055 \pm 0.002$ | $1.138 \pm 0.014$ | $0.336 \pm 0.005$ |
| Snapshot Ensemble | $77.0 \pm 0.1$ | $77.2 \pm 0.2$ | $0.123 \pm 0.002$ | $4.416 \pm 0.046$ | $1.614 \pm 0.007$ |
| Explicit Ensemble | $\underline{79.8} \pm 0.1$ | $\underline{79.8} \pm 0.2$ | $0.100 \pm 0.001$ | $0.745 \pm 0.003$ | $\underline{0.284} \pm 0.002$ |
| LoRA-Ensemble | $\mathbf{82.5} \pm 0.1$ | $\mathbf{82.5} \pm 0.1$ | $\underline{0.035} \pm 0.001$ | $\mathbf{0.587} \pm 0.001$ | $\mathbf{0.253} \pm 0.000$ |

Table 3: Model performance on the HAM10000 dataset for the compared methods. Ensembles have 16 members. Best score for each metric in **bold**, second-best underlined
.

| Method | Accuracy ($\uparrow$) | F1 ($\uparrow$) | ECE ($\downarrow$) | NLL ($\downarrow$) | Brier ($\downarrow$) |
|---|---|---|---|---|---|
| Single Network | $84.1 \pm 0.3$ | $71.4 \pm 0.7$ | $0.139 \pm 0.004$ | $1.138 \pm 0.040$ | $0.291 \pm 0.009$ |
| Single Net w/ LoRA | $83.2 \pm 0.7$ | $70.7 \pm 1.3$ | $0.085 \pm 0.004$ | $0.569 \pm 0.027$ | $0.256 \pm 0.011$ |
| MC Dropout | $83.7 \pm 0.4$ | $71.0 \pm 0.9$ | $0.099 \pm 0.007$ | $0.631 \pm 0.023$ | $0.270 \pm 0.009$ |
| Snapshot Ensemble | $84.9 \pm 0.3$ | $73.7 \pm 0.9$ | $\underline{0.058} \pm 0.004$ | $\underline{0.431} \pm 0.007$ | $\underline{0.217} \pm 0.004$ |
| Explicit Ensemble | $\underline{85.8} \pm 0.2$ | $\underline{74.6} \pm 0.4$ | $0.105 \pm 0.002$ | $0.536 \pm 0.007$ | $0.218 \pm 0.002$ |
| LoRA-Ensemble | $\mathbf{88.0} \pm 0.2$ | $\mathbf{78.3} \pm 0.6$ | $\mathbf{0.037} \pm 0.002$ | $\mathbf{0.342} \pm 0.003$ | $\mathbf{0.175} \pm 0.002$ |

creating an ensemble. This effect is not present when looking at NLL and Brier score, though, Tab 2. The reliability diagram in Fig. 5 in the Appendix somewhat elucidates this unexpected behavior. It turns out that LoRA-Ensemble is under-confident on CIFAR-100, meaning that the classification is more accurate than the model suggests. Rahaman & Thiery (2020) have found that when ensembling under-confident models, the accuracy grows faster than the confidence. As a result, the difference between accuracy and confidence tends to grow, worsening calibration metrics. Note that in safety-critical applications under-confident models that over-estimate the uncertainty are often preferable to over-confident ones.

MC Dropout is not well calibrated for smaller ensembles, but progressively catches up as the ensemble size increases. Snapshot Ensemble performs similarly to MC Dropout in terms of accuracy but does not perform competitively for calibration.

## 3.3 HAM10000 LESION CLASSIFICATION

In many medical applications, well-calibrated models are essential. As a test case, we use the classification of pigmented skin lesions and again compare the same group of models in terms of accuracy and calibration. The results are summarized in Tab. 3.

Similar to the CIFAR-100 evaluation, LoRA-Ensemble outperforms all other methods by a clear margin, with respect to both classification accuracy and calibration. Surprisingly, Snapshot Ensemble performs very well in terms of calibration but is not competitive as far as accuracy is concerned. The experiments also further support the above discussion of confidence vs. ensemble size (Sec. 3.2). For HAM10000 LoRA-Ensemble is slightly over-confident (just like the Explicit Ensemble) and, indeed, its calibration error decreases with ensemble size in this case, see Appendix A.3.

We conducted further experiments on HAM10000 using different backbone architectures with varying numbers of parameters. The results are shown in Tab. 8 in Appendix B. In conclusion, LoRA-Ensemble generalizes effectively across different backbones. Moreover, as the number of parameters in the backbone architecture increases, the superiority of LoRA-Ensemble over Explicit Ensemble in both accuracy and calibration becomes more pronounced.

Table 4: Model performance on the ESC-50 dataset for the compared methods. Ensembles have 8 members due to memory limitations. Best score for each metric in **bold**, second-best underlined.

| Method | Accuracy (↑) | F1 (↑) | ECE (↓) | NLL (↓) | Brier (↓) |
|---|---|---|---|---|---|
| Single Network | $89.6 \pm 0.7$ | $89.5 \pm 0.7$ | $0.039 \pm 0.004$ | $0.410 \pm 0.020$ | $0.164 \pm 0.009$ |
| Single Net w/ LoRA | $88.0 \pm 0.3$ | $87.8 \pm 0.3$ | $0.043 \pm 0.004$ | $0.461 \pm 0.019$ | $0.186 \pm 0.005$ |
| MC Dropout | $89.4 \pm 0.3$ | $89.3 \pm 0.4$ | $0.087 \pm 0.005$ | $0.553 \pm 0.012$ | $0.176 \pm 0.005$ |
| Explicit Ensemble | $\mathbf{91.3} \pm 0.2$ | $\mathbf{91.2} \pm 0.3$ | $\underline{0.027} \pm 0.004$ | $\mathbf{0.322} \pm 0.004$ | $\mathbf{0.133} \pm 0.001$ |
| LoRA-Ensemble | $\underline{91.1} \pm 0.2$ | $\underline{90.8} \pm 0.2$ | $\mathbf{0.021} \pm 0.003$ | $\underline{0.328} \pm 0.004$ | $\underline{0.138} \pm 0.001$ |

Table 5: Model performance on the OOD task. CIFAR-100 is used as the in-distribution dataset and CIFAR-10 and SVHN as the out-of-distribution dataset. Ensembles for all methods consist of 16 members. Results for Split-Ensemble are taken from Chen et al. (2024). The best score for each metric is highlighted in **bold**, with the second-best score underlined.

| OOD Dataset | CIFAR-10 | | SVHN | |
|---|---|---|---|---|
| Method | AUROC (↑) | AUPRC (↑) | AUROC (↑) | AUPRC (↑) |
| Split-Ensemble Chen et al. (2024) | 79.2 | 81.7 | 81.2 | 69.9 |
| Single Network | $75.6 \pm 0.3$ | $77.6 \pm 0.6$ | $76.4 \pm 1.8$ | $67.1 \pm 2.3$ |
| Single Network with LoRA | $\underline{80.1} \pm 0.5$ | $\underline{82.4} \pm 0.6$ | $\underline{85.9} \pm 0.9$ | $\underline{75.4} \pm 1.7$ |
| MC Dropout | $75.1 \pm 0.5$ | $73.7 \pm 0.9$ | $52.3 \pm 12.4$ | $29.9 \pm 7.1$ |
| Explicit Ensemble | $78.9 \pm 0.2$ | $80.8 \pm 0.2$ | $74.8 \pm 1.3$ | $63.9 \pm 1.5$ |
| LoRA-Ensemble | $\mathbf{82.1} \pm 0.1$ | $\mathbf{84.1} \pm 0.1$ | $\mathbf{89.9} \pm 0.6$ | $\mathbf{80.9} \pm 1.0$ |

### 3.4 ESC-50 Environmental Sound Classification

To go beyond computer vision tasks, LoRA-Ensemble is also applied to an audio dataset, using the Audio Spectrogram Transformer as the backbone model. The results are summarized in Tab. 4. On this dataset LoRA-Ensemble does not significantly outperform the Explicit Ensemble, but still matches its performance with much lower computational demands, see Appendix L. Accuracy is insignificantly lower, whereas calibration is slightly better in terms of ECE. We note that, remarkably, the weights used in the transformer modules and for creating patch embeddings were pre-trained on images rather than audio streams.

### 3.5 Out-of-Distribution (OOD) Detection

To evaluate our method's effectiveness in OOD detection, a crucial aspect of quantifying uncertainty in deep learning models Hendrycks & Gimpel (2016), we conducted an experiment where models were trained on CIFAR-100 (in-distribution) and tested on samples from both CIFAR-100 and CIFAR-10 or SVHN (out-of-distribution). Following Sim et al. (2023) and Chen et al. (2024), we used the maximum softmax probability as the confidence score. Tab. 5 highlights that the LoRA-Ensemble achieves superior performance compared to all other methods across both settings and metrics, surpassing even the recently proposed Split-Ensemble approach Chen et al. (2024), which was specifically designed for OOD tasks. Furthermore, consistent with our earlier observations on LoRA's effectiveness in improving network calibration, even a single LoRA model outperforms the Explicit Ensemble, highlighting its robustness in OOD scenarios.

## 4 Enhanced Diversity in LoRA-Ensemble

This section explores the diversity of ensemble members in function and weight space for LoRA-Ensemble and Explicit Ensemble, using the HAM10000 dataset with 16 ensemble members. Diversity is crucial for effective ensembles, as highly correlated members offer limited value (Zhang, 2012). By capturing diverse parameter configurations that equally explain the observations, ensembles can provide a more comprehensive quantification of epistemic uncertainty (Kendall & Gal, 2017).

Following Fort et al. (2019b), we first assess function space diversity through the predictions of individual ensemble members. In Fig. 2, we first compute the disagreement rate on the test set, defined as $\frac{1}{N} \sum_{n=1}^{N} \mathbb{I}[T_{\theta_i}(X_n) \neq T_{\theta_j}(X_n)]$, where $T_{\theta_i}(X_n)$ represents the class label predicted by ensemble member $i$ for input $X_n$, and $\mathbb{I}$ is the indicator function. Next, we construct a probability distribution for each ensemble member by aggregating their softmax outputs across all test samples and compute pairwise Jensen-Shannon divergences (JSD). Finally, we use t-SNE (Van der Maaten & Hinton, 2008) to visualize their spread in function space (aggregated softmax outputs). The analysis reveals that LoRA-Ensemble exhibits significantly higher diversity among ensemble members compared to an Explicit Ensemble. I.e., LoRA-Ensemble appears to capture a wider range of modes in function space compared to the Explicit Ensemble.

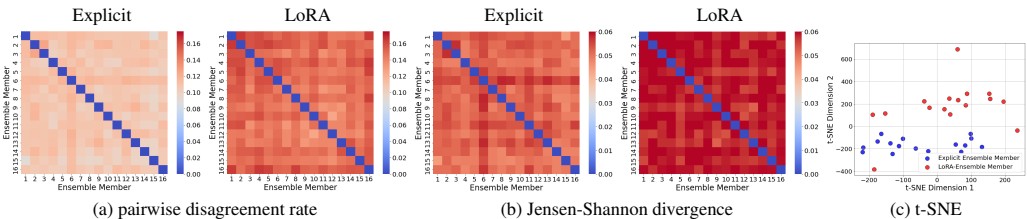

Figure 2: Function space analysis of LoRA-Ensemble vs. Explicit Ensemble.

We further inspect the weight spaces of LoRA-Ensemble and Explicit Ensemble with spectral analysis, focusing on the projection matrices within the attention blocks of the ViT (Base-32) model pre-trained on ImageNet. We show the analysis for value projection matrices, given their strong association with learned representations; details for query and key projection matrices are provided in Appendix D. We use Singular Value Decomposition (SVD) to identify the most significant transformations encoded in the weights, as larger singular values capture the most impactful components.

Following Shuttleworth et al. (2024), we analyze the similarity between the initial (pre-trained) weights and the final trained weights of ensemble members. LoRA-Ensemble and Explicit Ensemble lead to very different parameter updates. LoRA-Ensemble introduces new high-ranking singular vectors that are near-orthogonal to those in the initial weights, referred to as "intruder dimensions" (Shuttleworth et al., 2024). In contrast, Explicit Ensemble members tend to adhere closely to the spectral structure of the initial weights (see Fig. 10 in Appendix).

The random initialization of matrices $A$ and $B$ in the LoRA module leads to an intriguing phenomenon: the intruder dimensions of different LoRA-Ensemble members are near-orthogonal, see cosine similarities between the highest-ranking singular vectors of different members in Fig. 3 (for details see Appendix D). The rank is set to 4 and similarities are averaged over layers and pairs of members. Notably, the highest-ranked singular vectors of distinct members exhibit almost no similarity; in contrast to the Explicit Ensemble, where they are highly correlated. The weight-space cosine similarity provides further evidence of enhanced diversity. LoRA-Ensemble members exhibit greatly increased diversity in weight space. To visualize training trajectories, we apply t-SNE to plot the evolution of weights during training for both methods. LoRA-Ensemble members converge across a broader area of the loss landscape, indicating diverse learning dynamics. In contrast, Explicit Ensemble members remain closer to the initial weights, reflecting reduced diversity. Overall, these results highlight that the LoRA-Ensemble better explores the weight space, and thus the epistemic uncertainty.

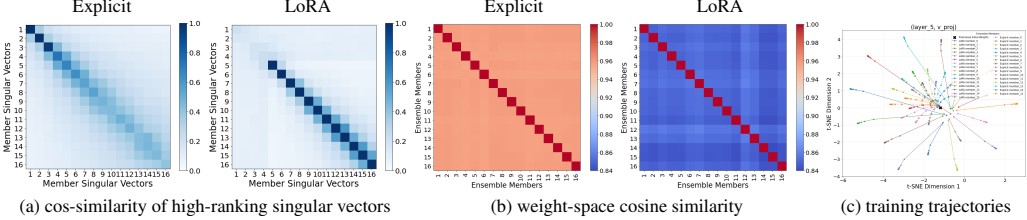

Figure 3: Weight space analysis of LoRA-Ensemble vs. Explicit Ensemble.

## 5 RELATED WORK

**Estimation of Epistemic Uncertainty**   A lot of work has gone into estimating the epistemic uncertainty in Artificial Neural Network (ANN). As the analytical computation of the posterior in such models is generally intractable, methods for approximate Bayesian inference have been proposed. Such methods rely on imposing an appropriate prior on the weights and using the likelihood of the training data to get an approximate posterior of the weight space.

The main techniques are, on the one hand, Variational Inference (Graves, 2011; Ranganath et al., 2014), which Blundell et al. (2015) have specialized to neural networks as *Bayes by Backprop*. And on the other hand variants of Markov Chain Monte Carlo (MCMC) (Neal, 1996; Chen et al., 2014), including Stochastic Gradient Langevin Dynamics (SGLD) (Welling et al., 2011). These, however, are often not able to accurately capture high-dimensional and highly non-convex loss landscapes, like the ones usually encountered in deep learning (Gustafsson et al., 2019).

**Ensembles and Implicit Ensembling**   Lakshminarayanan et al. (2017) have proposed a method known as deep ensembles. It uses a set of neural networks with identical architecture that are independently and randomly initialized, and (as usual) trained with variants of Stochastic Gradient Descent (SGD). While the latter introduces further stochasticity, Fort et al. (2019a) have shown that the initialization of the weights is more important to explore the admissible weight space. Ensemble members will generally converge to different modes of the loss function, such that they can be considered Monte Carlo samples of the posterior distribution (Wilson & Izmailov, 2020; Izmailov et al., 2021). While ensembles, in general, yield the best results in terms of accuracy and uncertainty calibration, a straightforward implementation suffers from high memory and compute requirements, since multiple instances of the full neural network must be trained and stored. This can become prohibitive for modern neural networks with many millions, or even billions, of parameters.

Consequently, researchers have attempted to find ways of mimicking the principle of deep ensembles without creating several full copies of the base model. Gal & Ghahramani (2015) have proposed Monte Carlo Dropout, where the posterior is approximated by sampling different dropout patterns at inference time. While this is less expensive in terms of memory, performance is often worse. Masksembles (Durasov et al., 2020) are a variant that attempts to select suitable dropout masks in order to obtain better uncertainty estimates. Snapshot Ensembles (Huang et al., 2017) use cyclic learning rates to steer the learning process such that it passes through multiple local minima, which are then stored as ensemble members. This reduces the training effort but does not address memory requirements or inference time.

Particularly relevant for our work are attempts that employ a shared backbone and modify only selected layers. Havasi et al. (2020) follow that strategy, in their case only the first and last layer of a neural network are replicated and trained independently to emulate an ensemble. Packed-Ensemble (Laurent et al., 2023) leverage grouped convolutions to train lightweight ensembles within a single shared backbone. BatchEnsemble (Wen et al., 2020) is similar to LoRA-Ensemble in that it also uses low-rank matrices to change the model parameters. More specifically, shared weight matrices are modulated by element-wise multiplication with different rank-1 matrices to achieve the behavior of a deep ensemble while adding only a small number of parameters. Wenzel et al. (2020) take this concept further by also ensembling over different hyper-parameter settings. Turkoglu et al. (2022) freeze all weights of the base model and instead vary the feature-wise linear modulation (FiLM, Li et al., 2018b; Takeda et al., 2021).

A related concept was recently introduced for LLMs: the Mixtral of Experts model (Jiang et al., 2024) averages over a sparse mixture of experts to efficiently generate text.

**Low-Rank Adaptation in Transformer Networks**   Low-Rank Adaptation was originally conceived as a parameter-efficient way of fine-tuning Large Language Models (LLMs) (Hu et al., 2021). It is based on the observation that, while modern neural networks have huge parameter spaces, the solutions they converge to have much lower intrinsic dimension (Li et al., 2018b; Aghajanyan et al., 2020). LoRA exploits this and Hu et al. (2021) show that even when fine-tuning only low-rank update matrix $B \cdot A$ (sometimes with rank as low as one or two), the resulting models are competitive with much more expensive fine-tuning schemes. The method quickly became popular and has since also been extended with weight-decomposition (Liu et al., 2024). The Low-Rank Adaptation (LoRA) idea

has been applied in various fields, notably for denoising diffusion models (Luo et al., 2023; Golnari, 2023).

As we have shown, LoRA's adaptation technique naturally lends itself to parameter-efficient ensembling. We study the resulting ensemble for uncertainty calibration, a similar approach has concurrently been explored for the purpose of fine-tuning large language models (Wang et al., 2023) with promising results.

## 6 DISCUSSION & CONCLUSION

**On Effectiveness of LoRA-Ensemble** Across diverse tasks, our experiments consistently show that LoRA-Ensemble matches or surpasses the predictive performance of the state-of-the-art Explicit Ensemble while offering superior calibration. Adding LoRA to a single model *without any ensembling* improves calibration in most experiments beyond that of a 16-member Explicit Ensemble. This effect may be linked to the well-documented over-parameterization of modern neural networks, which often achieve higher predictive accuracy at the cost of poorer calibration (e.g., Guo et al., 2017). By incorporating LoRA while treating all pre-trained weights as constants, we significantly reduce the trainable parameter space, potentially favoring better calibration. However, limiting trainable parameters alone does not ensure better accuracy or calibration, as forms of regularization or selective training may fall short. The effectiveness of LoRA-Ensemble stems from its unique learning dynamics, which we explore in Sec. 4 and Appendix D. LoRA-Ensemble members converge across a broader area of the loss landscape, enabling better exploration of the weight space and more effective capture of epistemic uncertainty. Increasing the number of members in the LoRA-Ensemble enhances predictive power, potentially improving accuracy while maintaining good calibration due to the limited number of trainable weights. However, if the trainable weights are not kept limited (e.g., by increasing the LoRA rank), calibration can worsen, as demonstrated in Fig.8a and Tab.7. This effect aligns with findings by Shuttleworth et al. (2024), which indicate that excessively increasing the rank in LoRA can cause it to lose its unique learning dynamics. Conversely, enhancing predictive power by increasing the pre-trained weights (while keeping trainable weights constant) further improves the effectiveness of the LoRA-Ensemble, see Appendix B.

**Limitations** We propose a parameter-efficient ensembling method that performs well in the conducted experiments. While we did not evaluate LoRA-Ensemble on very large datasets, such as those often found in natural language processing, it would be interesting to explore its performance on such datasets. Similarly, evaluating its effectiveness on Large Language Models would be valuable, given their increasing popularity. Although our method addresses the restrictive memory usage of traditional ensembles, it does not reduce computational complexity, as data must still pass through the model once per batch. Additionally, approximate inference on the parameter distribution of the LoRA matrices could allow drawing an infinite number of ensemble members from the approximate posterior.

**Future Work** As discussed by Rahaman & Thiery (2020), our work also suggests that in a high-parameter regime, deep ensembles may not exhibit the same behavior as they do in a low-parameter regime, where they typically improve calibration properties. We have previously witnessed this type of phase shift in bias-variance trade-off for large neural networks akin to Double Descent Phenomena (Nakkiran et al., 2021). It would be valuable to conduct an in-depth analysis of deep ensemble behavior in high-parameter regimes, while also considering data size, model size, and compute.

**Conclusion** We have presented LoRA-Ensemble, a novel, parameter-efficient method for probabilistic learning that is tailored to the transformer architecture (and potentially other architectures that make use of the attention mechanism). LoRA-Ensemble uses a simple, but efficient trick to turn a single base model into an implicit ensemble: the weights of the base model are kept frozen, but are modulated with the Low-Rank Adaptation mechanism. By training multiple, stochastically varying instances of the low-rank matrices that define the modulation, one obtains a diverse set of ensemble members that share the majority of their weights (specifically, those of the base model) and introduces only minimal overhead through the coefficients of their individual low-rank matrices. Our experiments on two different computer vision tasks, a sound classification task, and an OOD detection task show that the proposed approach can outperform other, implicit as well as explicit, ensembling strategies in terms of both classification performance and uncertainty calibration.

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

## A  MORE EXPERIMENTS & RESULTS

This section presents comprehensive experimental results for a new dataset: CIFAR-10 and includes additional figures for the HAM10000 dataset.

### A.1  CIFAR-10

The results for the CIFAR-10 dataset, as shown in Tab. 6, indicate that LoRA-Ensemble outperforms all other methods across all metrics. Following closely is a single network enhanced with LoRA. This mirrors the results found in the main paper for CIFAR-100, with the exception of the calibration for a single model. It is important to note that although all methods achieve high accuracy and the differences between them are minimal, calibration is nearly perfect for most approaches. This suggests that the CIFAR-10 dataset is relatively easy for modern transformer models, and the results should not be over-interpreted. Nevertheless, the consistent performance across different random seeds suggests that the ranking is likely significant. Given the balanced nature of the CIFAR-10 dataset, the accuracy and F1-score are almost identical.

Table 6: Performance on the CIFAR-10 dataset for all compared methods. Ensembles have 16 members. Best score for each metric in **bold**, second-best underlined.

| Method | Accuracy (↑) | F1 (↑) | ECE (↓) | NLL (↓) | Brier (↓) |
|---|---|---|---|---|---|
| Single Network | $92.8 \pm 0.1$ | $92.8 \pm 0.1$ | $0.051 \pm 0.001$ | $0.333 \pm 0.003$ | $0.120 \pm 0.002$ |
| Single Net w/ LoRA | $\underline{94.5} \pm 0.0$ | $\underline{94.5} \pm 0.0$ | $\underline{0.009} \pm 0.001$ | $\underline{0.163} \pm 0.002$ | $\underline{0.082} \pm 0.001$ |
| MC Dropout | $92.9 \pm 0.2$ | $92.9 \pm 0.2$ | $0.023 \pm 0.002$ | $0.260 \pm 0.005$ | $0.110 \pm 0.003$ |
| Explicit Ensemble | $94.1 \pm 0.1$ | $94.1 \pm 0.1$ | $0.031 \pm 0.001$ | $0.181 \pm 0.002$ | $0.087 \pm 0.001$ |
| Snapshot Ensemble | $93.1 \pm 0.1$ | $93.1 \pm 0.1$ | $0.037 \pm 0.002$ | $1.062 \pm 0.021$ | $0.510 \pm 0.008$ |
| LoRA-Ensemble | $\mathbf{95.9} \pm 0.1$ | $\mathbf{95.9} \pm 0.1$ | $\mathbf{0.003} \pm 0.001$ | $\mathbf{0.128} \pm 0.001$ | $\mathbf{0.064} \pm 0.000$ |

### A.2  CIFAR-100

Increasing the ensemble size of LoRA-Ensemble on CIFAR-100 improves classification accuracy but reduces calibration, as illustrated in Fig. 4. The reliability diagram in Fig. 5 highlights this behavior: networks with LoRA on CIFAR-100 are generally under-confident, with accuracy exceeding predicted confidence. As observed by Rahaman & Thiery (2020), ensembling under-confident models can exacerbate this discrepancy, leading to poorer calibration metrics.

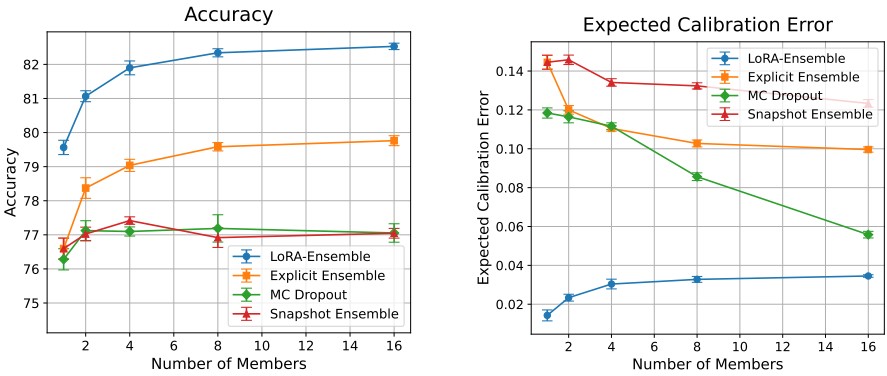

Figure 4: Accuracy and Expected Calibration Error on CIFAR-100, with different ensemble sizes.

### A.3  HAM10000 LESION CLASSIFICATION

Classification accuracy and ECE for HAM10000 dataset are both graphed against ensemble size in Fig. 6. Again, LoRA-Ensemble outperforms all baselines for larger ensembles. In Fig. 7 the reliability

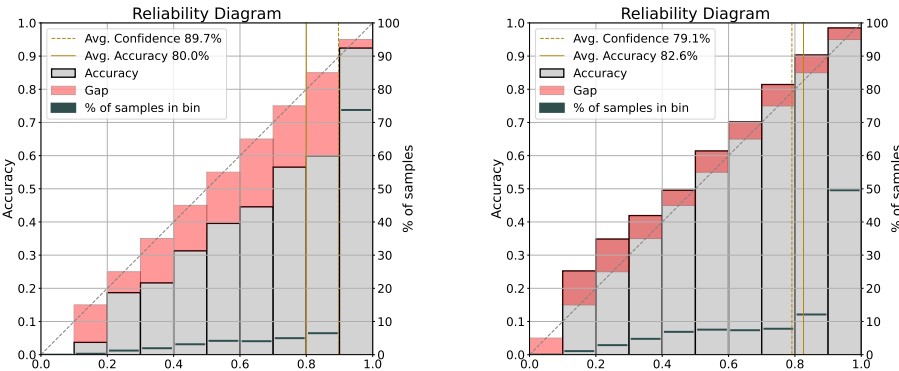

Figure 5: Reliability diagrams for Explicit Ensemble (left) and LoRA-Ensemble (right) with 16 members, on CIFAR-100.

diagrams for LoRA-Ensemble and an Explicit Ensemble with 16 members each on the HAM10000 dataset are shown. Here, the models are overconfident, further supporting our reasoning regarding the surprising behaviour of calibration with growing ensemble size in the case of CIFAR-100.

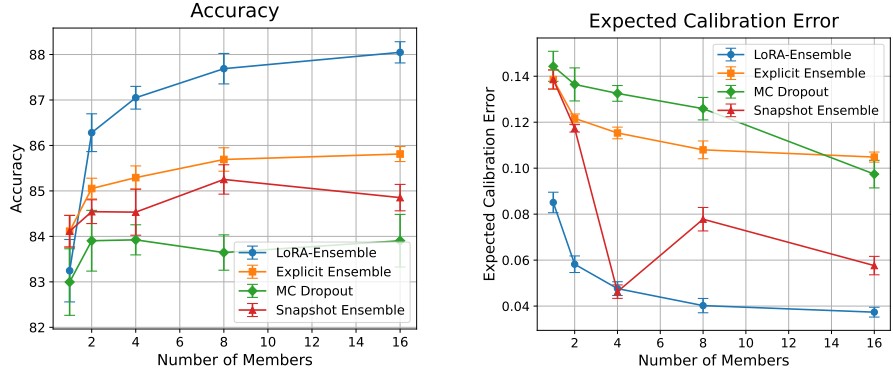

Figure 6: Accuracy and Expected Calibration Error on HAM10000, with different ensemble sizes.

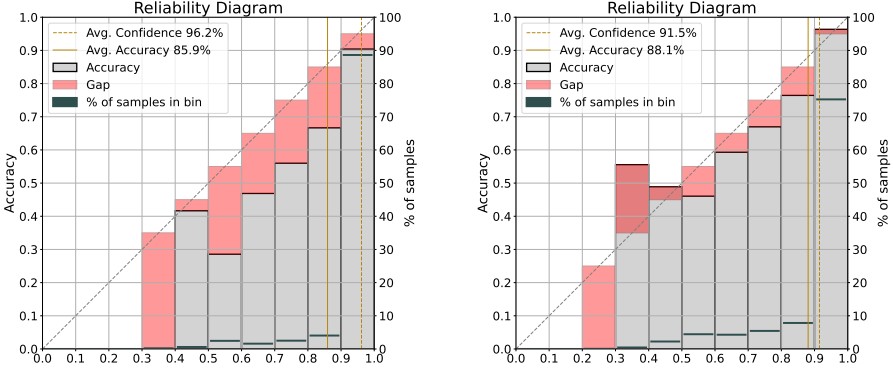

Figure 7: Reliability diagrams for Explicit Ensemble (left) and LoRA-Ensemble (right) with 16 members, on HAM10000.

### A.4 MORE BASELINE COMPARISON

We compare the proposed LoRA-Ensemble method with an additional baseline: a single high-rank LoRA model configured to have the same total number of trainable LoRA parameters as the LoRA-

Table 7: Model performance on the CIFAR-100 dataset for the compared methods. Ensembles have 16 members. Best score for each metric in **bold**, second-best underlined.

| Method | Rank | Trainable params. | Accuracy (↑) | F1 (↑) | ECE (↓) | NLL (↓) | Brier (↓) |
|---|---|---|---|---|---|---|---|
| Single Net w/ LoRA | 8 | 666'724 | 79.6 ± 0.2 | 79.4 ± 0.2 | **0.014** ± 0.003 | 0.671 ± 0.005 | 0.286 ± 0.003 |
| Single Net w/ LoRA | 128 | 9'514'084 | 77.0 ± 0.1 | 77.0 ± 0.1 | 0.080 ± 0.001 | 0.867 ± 0.007 | 0.332 ± 0.002 |
| LoRA-Ensemble | 8 | 10'667'584 | **82.5** ± 0.1 | **82.5** ± 0.1 | 0.035 ± 0.001 | **0.587** ± 0.001 | **0.253** ± 0.000 |

Ensemble. This evaluation is conducted on the CIFAR-100 classification task to examine the relative effectiveness of ensembling versus increasing parameter capacity within a single model.

Notably, as shown in Tab. 7, the high-rank LoRA model underperforms compared to the low-rank LoRA model. This result indicates that the performance gains of the LoRA-Ensemble are not solely due to an increased number of trainable parameters but are instead attributable to the ensembling approach.

# B  EFFECT OF MODEL SIZE ON PREDICTION AND CALIBRATION PERFORMANCE

Building upon our existing experiments with the HAM10000 dataset, we extended our analysis to include different backbone architectures with varying numbers of parameters. Specifically, we utilized various DeiT models pre-trained with distillation, as described by Touvron et al. (2020). The results are presented in Table 8. Notably, the DeiT Base-32 model is the same as the ViT Base-32 model.

In the small parameter regime (Tiny-16, Small-16), the addition of a single LoRA module did not consistently enhance calibration compared to using a single model. This observation contrasts with our findings in most other experiments. However, in the larger parameter regime (ViT Base-32), incorporating even a single LoRA module significantly improved calibration.

Furthermore, increasing the number of ensembles in the LoRA-Ensemble not only boosted accuracy but also enhanced calibration, enabling it to match the performance of an Explicit Ensemble in both parameter regimes. Finally, as the number of parameters in the backbone architecture increased, the superiority of the LoRA-Ensemble over the Explicit Ensemble in terms of both accuracy and calibration became more pronounced.

Table 8: Performance metrics on the HAM10000 dataset for different Vision Transformer architectures. Ensembles have 16 members. Best score for each metric in **bold**, second-best underlined.

| Arch. | Method | # Params. | Accuracy (↑) | F1 (↑) | ECE (↓) | NLL (↓) | Brier (↓) |
|---|---|---|---|---|---|---|---|
| DeiT Tiny-16 | Single Net | 5 M | 89.0 ± 0.3 | 79.0 ± 0.4 | 0.096 ± 0.003 | 0.909 ± 0.037 | 0.202 ± 0.005 |
| | Single Net w/ LoRA | | 84.5 ± 0.8 | 71.6 ± 1.5 | 0.074 ± 0.003 | 0.542 ± 0.017 | 0.237 ± 0.009 |
| | Explicit Ensemble | | **90.4** ± 0.3 | **81.4** ± 0.4 | 0.069 ± 0.004 | 0.340 ± 0.006 | **0.142** ± 0.002 |
| | LoRA-Ensemble | | 88.9 ± 0.4 | 80.6 ± 0.2 | **0.025** ± 0.003 | **0.325** ± 0.004 | 0.164 ± 0.002 |
| DeiT Small-16 | Single Net | 22 M | 89.6 ± 0.4 | 79.0 ± 0.5 | 0.093 ± 0.003 | 0.876 ± 0.032 | 0.191 ± 0.007 |
| | Single Net w/ LoRA | | 86.3 ± 0.5 | 76.8 ± 1.0 | 0.100 ± 0.007 | 0.731 ± 0.053 | 0.234 ± 0.010 |
| | Explicit Ensemble | | **91.5** ± 0.1 | 82.4 ± 0.2 | 0.061 ± 0.003 | 0.318 ± 0.003 | **0.130** ± 0.001 |
| | LoRA-Ensemble | | 90.4 ± 0.1 | **82.8** ± 0.4 | **0.047** ± 0.002 | **0.292** ± 0.002 | 0.144 ± 0.001 |
| DeiT Base-32 | Single Net | 86 M | 84.1 ± 0.3 | 71.4 ± 0.7 | 0.139 ± 0.004 | 1.138 ± 0.040 | 0.291 ± 0.009 |
| | Single Net w/ LoRA | | 83.2 ± 0.7 | 70.7 ± 1.3 | 0.085 ± 0.004 | 0.569 ± 0.027 | 0.256 ± 0.011 |
| | Explicit Ensemble | | 85.8 ± 0.2 | 74.6 ± 0.4 | 0.105 ± 0.002 | 0.536 ± 0.007 | 0.218 ± 0.002 |
| | LoRA-Ensemble | | **88.0** ± 0.2 | **78.3** ± 0.6 | **0.037** ± 0.002 | **0.342** ± 0.003 | **0.175** ± 0.002 |

# C  SENSITIVITY ANALYSIS: LORA RANK

The main hyper-parameter introduced by adding LoRA is the rank of the low-rank decomposition (i.e., the common dimension of the matrices $A$ and $B$). Varying that rank modulates the complexity of

the model for the learning task. We have empirically studied the relationship between rank, accuracy, and Expected Calibration Error. Here we show results for HAM10000 and CIFAR-100 dataset.

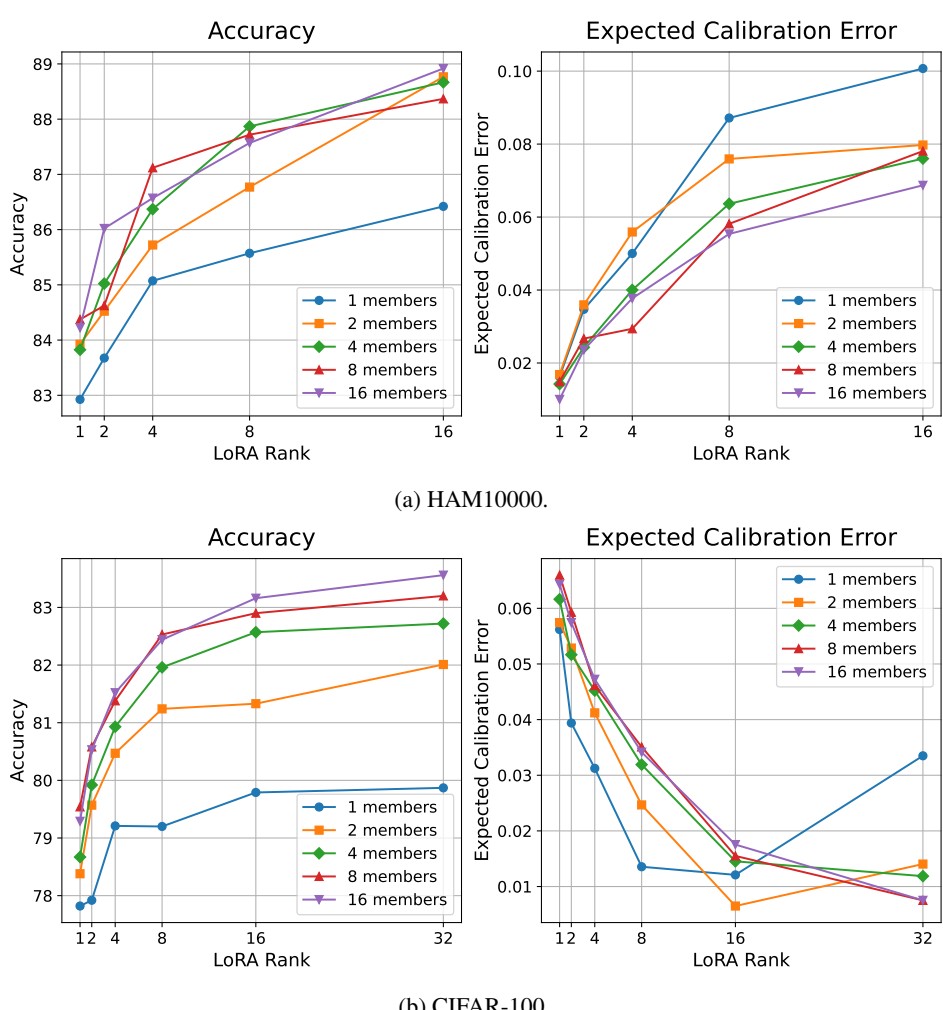

(a) HAM10000.

(b) CIFAR-100

Figure 8: Impact of LoRA rank on accuracy and ECE.

On HAM10000 we observe a clear trade-off between accuracy and calibration, Fig. 8a. With increasing rank the classification accuracy increases while the calibration deteriorates, in other words, one can to some degree balance predictive accuracy against uncertainty calibration by choosing the rank. Our focus in this work is on model calibration. We therefore generally choose the rank to favor calibration, even at the cost of slightly lower classification accuracy.

For the CIFAR-100 dataset, our evaluation of LoRA-Ensemble shows both increased accuracy and improved calibration with increasing rank within the studied range. These findings are illustrated in Fig. 8b.

This observation aligns with the findings of Rahaman & Thiery (2020), as LoRA-Ensemble continues to exhibit under-confidence even at higher ranks. Increasing model complexity enhances confidence, thereby improving calibration. However, at rank 32, the calibration of a single network augmented with LoRA begins to deteriorate, suggesting that a critical boundary has been reached. Beyond this point, the parameter space becomes insufficiently constrained, leading to effects similar to those observed by Guo et al. (2017).

At higher ranks, accuracy plateaus while memory demand increases linearly with $\mathcal{O}(d)$ and $\mathcal{O}(k)$ for $A \in \mathbb{R}^{r \times d}$ and $B \in \mathbb{R}^{k \times r}$ respectively, where $d$ and $k$ are the dimensions of the pre-trained weight matrix $W_0 \in \mathbb{R}^{k \times d}$. Consequently, we selected rank 8 for our CIFAR-100 experiments.

# D WEIGHT SPACE ANALYSIS

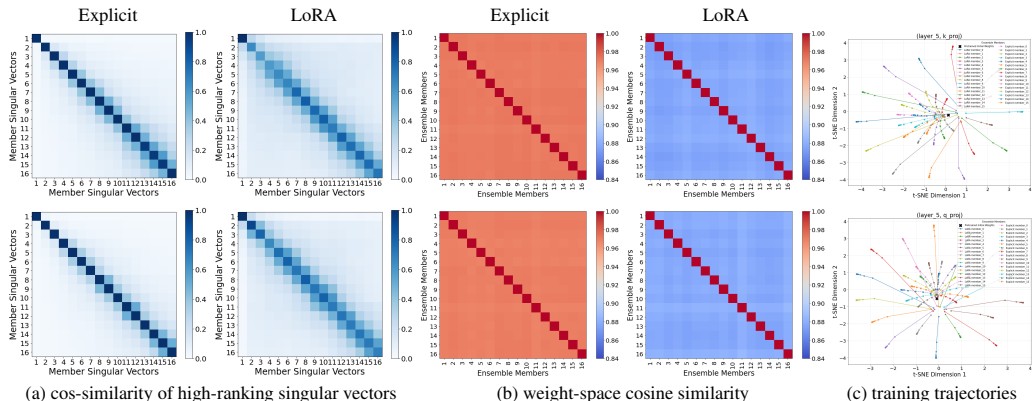

(a) cos-similarity of high-ranking singular vectors    (b) weight-space cosine similarity    (c) training trajectories

Figure 9: Weight space analysis of LoRA-Ensemble vs. Explicit Ensemble: The first row represents key matrices, while the second row represents query matrices.

This section expands on Sec. 4, which examines the diversity of ensemble members in function and weight space for LoRA-Ensemble and Explicit Ensemble, showing that LoRA-Ensemble exhibits greater diversity in both spaces. While Sec. 4 focuses on value projection matrices due to their role in learned representations, this section examines query and key projection matrices, too. In Fig. 9, we observe that LoRA-Ensemble achieves greater diversity in query and key projection matrices, similar to the diversity observed in value projection matrices (Fig. 3).

Using Singular Value Decomposition (SVD), a weight matrix $W \in \mathbb{R}^{m \times n}$ is decomposed as:

$$W = U \Sigma V^\top,$$

where $U \in \mathbb{R}^{m \times m}$ and $V \in \mathbb{R}^{n \times n}$ are orthonormal matrices representing rotational components, and $\Sigma \in \mathbb{R}^{m \times n}$ is a diagonal matrix of singular values capturing the scaling effect. Singular vectors linked to larger singular values highlight key transformations encoded by $W$.

In Fig. 10, we analyze the differences in weight updates between ensemble methods by computing the Singular Value Decomposition (SVD) of pre-trained and trained weights for ensemble members. Singular vectors corresponding to the top singular values (16 are shown) are extracted and compared using cosine similarity to evaluate changes in the weight structure. These similarities are averaged across layers and ensemble members. The results highlight distinct parameter update patterns between LoRA-Ensemble and Explicit Ensemble. LoRA-Ensemble introduces new high-ranking singular vectors, referred to as "intruder dimensions" (Shuttleworth et al., 2024), which are nearly orthogonal to the singular vectors of the pre-trained weights. The number of intruder dimensions depends on the LoRA rank. This effect is particularly pronounced in the value projection matrices, which aligns with their strong association with learned representations. In contrast, Explicit Ensemble members tend to preserve a structure closely aligned with the spectral properties of the pre-trained weights. This alignment is especially evident in the key and query projection matrices, which exhibit a strong resemblance to the original spectral structure.

We further analyze the $B \cdot A$ matrices learned by different ensemble members. Due to their random initialization, these matrices explore diverse directions in weight space. In Fig. 11, we plot the largest eigenvalues of these matrices (with only four non-zero eigenvalues as the LoRA rank is set to 4) and the similarity between the corresponding eigenvectors across ensemble members. The similarities are averaged over layers and member pairs. The results show that while the eigenvalues across members follow a similar trend, the eigenvectors are largely uncorrelated. This indicates that ensemble members explore different regions of weight space while maintaining similar overall transformations. The shared eigenvalue trends suggest consistent semantic contributions across members, while the dissimilar eigenvectors highlight the diversity in their learned representations.

We plot the t-SNE visualizations for different layers in Fig. 12, capturing the evolution of weights during training. The visualizations include the initial pretrained weights, and for each ensemble

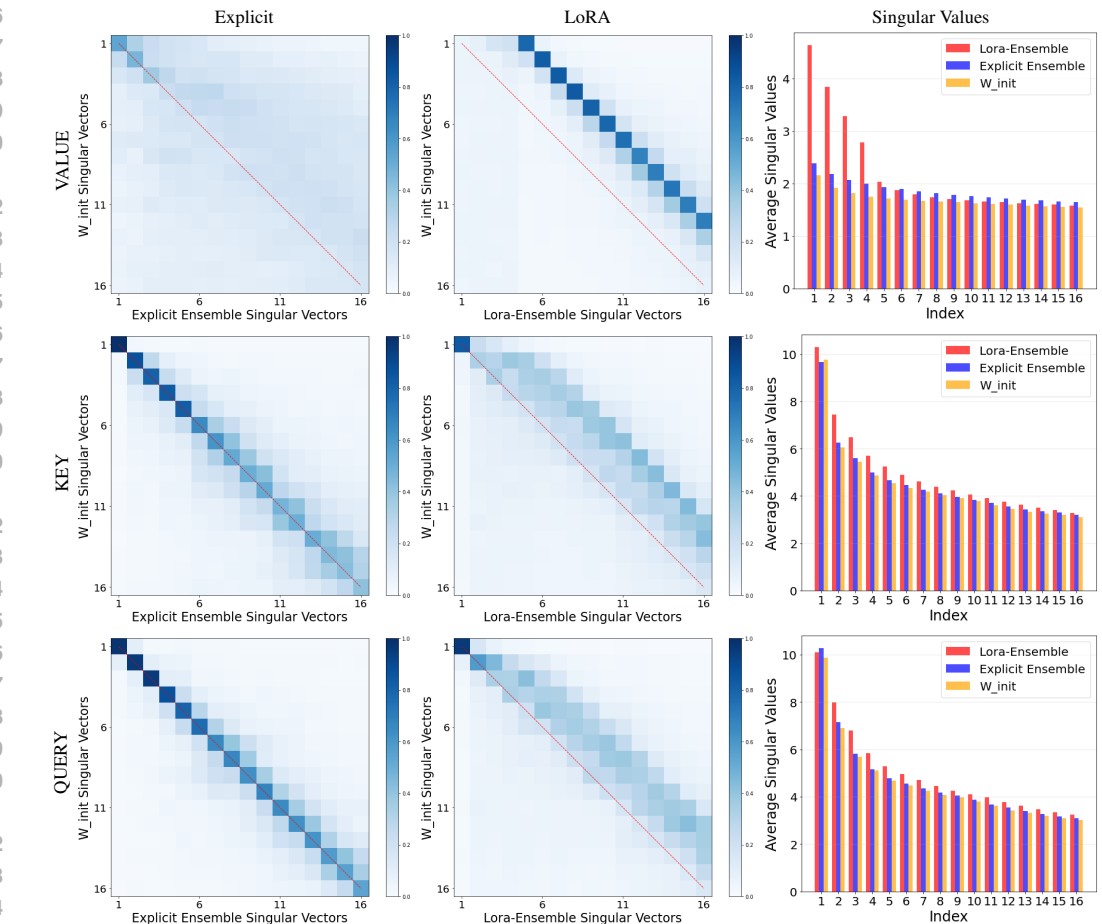

Figure 10: Cosine similarity of top singular vectors (and associated singular values) between initial pre-trained and final trained weights, averaged over layers and ensemble members.

member, we plot weights from epoch 5 to epoch 65 at 5-epoch intervals. The plots reveal that LoRA-Ensemble members exhibit broader convergence across the loss landscape in various layers, signifying diverse learning dynamics. Conversely, Explicit Ensemble members tend to remain closer to their initial weights, indicating reduced diversity throughout the training process.

## E    TEMPERATURE SCALING

Temperature scaling is a simple yet effective post-hoc calibration method used to improve the confidence of probabilistic models (Guo et al., 2017). It rescales the logits of a trained model by a scalar parameter $T > 0$ (the temperature). Given logits $\mathbf{z}$, the calibrated probabilities $\hat{p}_i$ for class $i$ are computed as:

$$\hat{p}_i = \frac{\exp(z_i/T)}{\sum_j \exp(z_j/T)}. \tag{5}$$

Here, $T = 1$ corresponds to no scaling, and $T > 1$ reduces overconfidence by softening the logits.

To assess the impact of temperature scaling on calibration, we conducted experiments on CIFAR-100 with varying temperature values, as shown in Tab. 9. For each method, the model parameters were fixed, and the effect of different temperatures on calibration was evaluated. We observe that calibration can be improved across all methods, with the exception of the single network with LoRA, which does not require temperature scaling.

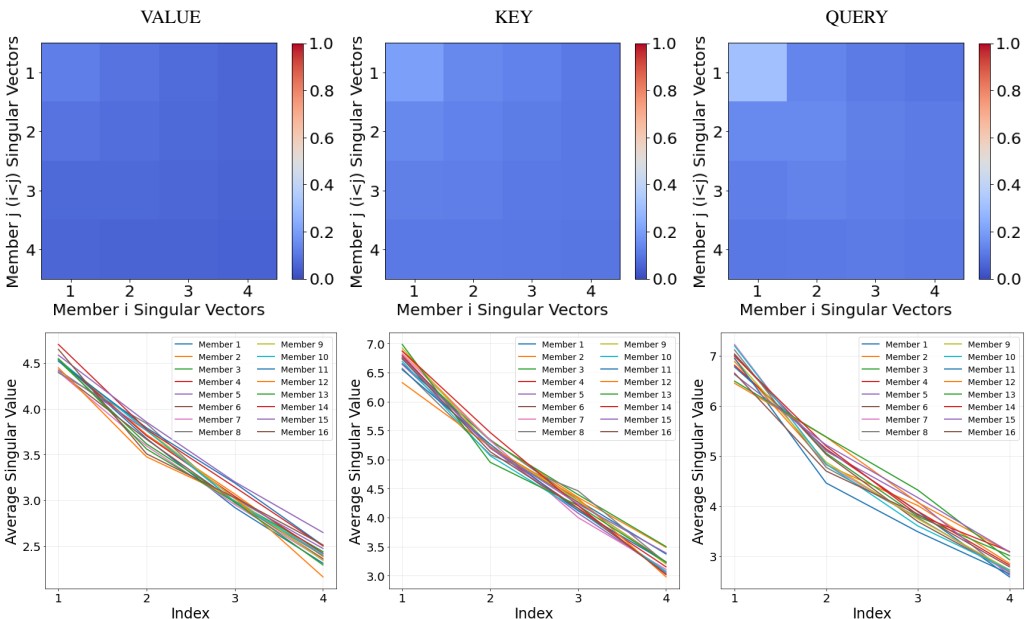

Figure 11: Cosine similarity of top singular vectors from $B \cdot A$ low-rank matrices (rank set to 4) between LoRA-Ensemble members, averaged over layers and all member pairs (first row), along with corresponding average singular values for different members (second row).

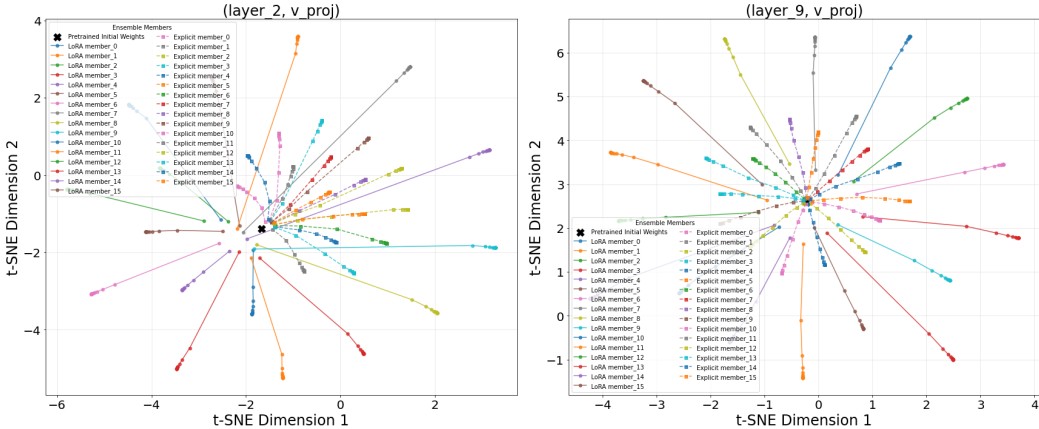

Figure 12: Training trajectories of ensemble members of LoRA-Ensemble and Explicit Ensemble.

As discussed in Section 3.2, LoRA-Ensemble is under-confident on CIFAR-100, as evidenced by the optimal temperature being less than 1 for this method.

## F  TRAINING DETAILS

The CIFAR-10/100 and HAM10000 dataset experiments are based on the ViT-Base-32 architecture (Dosovitskiy et al., 2020). This model has 12 layers and uses 768-dimensional patch embeddings, and the multi-head attention modules have 12 heads. All Vision Transformer models for image classification are trained using the AdamW optimizer (Loshchilov & Hutter, 2017). The base learning rate is initially set to 0.0001. The training uses a learning rate warm-up of 500 steps, where the learning rate increases linearly from 0 to the base learning rate before switching to a cosine decline over the rest of the steps. During the experiments, the gradients were calculated and then clipped not to exceed a maximum norm of 1. In the case of HAM10000, we used a weighted cross entropy

Table 9: Model performance on the CIFAR-100 dataset with different temperature. Best score for each metric and method in **bold**, second-best underlined.

| Method | Temp. | Accuracy ($\uparrow$) | F1 ($\uparrow$) | ECE ($\downarrow$) | NLL ($\downarrow$) | Brier ($\downarrow$) |
|---|---|---|---|---|---|---|
| Single Network | 1.4 | **76.8** | **76.7** | 0.091 | 0.969 | 0.344 |
| Single Network | 1.6 | | | 0.061 | 0.928 | 0.334 |
| Single Network | 1.8 | | | 0.034 | **0.920** | **0.329** |
| Single Network | 2.0 | | | **0.029** | 0.939 | **0.329** |
| Single Network | 2.2 | | | 0.078 | 0.982 | 0.335 |
| Single Net w/ LoRA | 0.4 | **79.2** | **79.1** | 0.130 | 1.020 | 0.332 |
| Single Net w/ LoRA | 0.6 | | | 0.088 | 0.772 | 0.308 |
| Single Net w/ LoRA | 0.8 | | | 0.042 | 0.688 | 0.294 |
| Single Net w/ LoRA | 1.0 | | | **0.013** | **0.680** | **0.290** |
| Single Net w/ LoRA | 1.2 | | | 0.073 | 0.722 | 0.298 |
| MC Dropout | 0.4 | **76.6** | **76.6** | 0.203 | 1.554 | 0.372 |
| MC Dropout | 0.6 | | | 0.174 | 1.223 | 0.361 |
| MC Dropout | 0.8 | | | 0.111 | **1.114** | 0.344 |
| MC Dropout | 1.0 | | | **0.057** | 1.163 | **0.342** |
| MC Dropout | 1.2 | | | 0.175 | 1.333 | 0.393 |
| Explicit Ensemble | 1.0 | **79.8** | **79.9** | 0.100 | 0.744 | 0.285 |
| Explicit Ensemble | 1.2 | | | 0.072 | 0.719 | 0.282 |
| Explicit Ensemble | 1.4 | | | 0.041 | **0.718** | **0.281** |
| Explicit Ensemble | 1.6 | | | **0.019** | 0.737 | 0.284 |
| Explicit Ensemble | 1.8 | | | 0.046 | 0.777 | 0.290 |
| LoRA-Ensemble | 0.4 | **82.4** | **82.4** | 0.103 | 0.628 | 0.252 |
| LoRA-Ensemble | 0.6 | | | 0.063 | 0.565 | **0.247** |
| LoRA-Ensemble | 0.8 | | | **0.018** | **0.557** | **0.247** |
| LoRA-Ensemble | 1.0 | | | 0.034 | 0.587 | 0.253 |
| LoRA-Ensemble | 1.2 | | | 0.095 | 0.650 | 0.269 |

loss that considered the estimated effective number of samples, which was determined using a beta parameter of 0.9991 (Cui et al., 2019). Uniform class weights were used for all other datasets. The maximum number of training epochs varies depending on the dataset. For CIFAR-100, the model is trained for 16 epochs (just over 25000 steps), while on HAM10000, it is trained for 65 epochs. Overall, the hyperparameters used in this work were loosely based on Conrad (2023). The models were trained using pre-trained weights from `torchvision 0.17.1` on an NVIDIA Tesla A100 graphics card. Moreover, the LoRA models were configured with a rank of 8 for both CIFAR-10 and CIFAR-100 and a rank of 4 for HAM10000. For Monte Carlo Dropout the dropout rate was empirically set to be $0.2$. Refer to Appendix M for details.

The settings used for the ESC-50 dataset training are similar to those used in Gong et al. (2021). However, we used a batch size of 1 instead of 48 to enable training on a single GPU. The base learning rate is set to 0.00001 for the Explicit Ensemble as well as MC Dropout experiments and 0.00005 for LoRA-Ensemble. These learning rates are lower than the ones used in Gong et al. (2021), which is due to the smaller batch size. Refer to the Appendix K for more details. The LoRA models were implemented with a rank of 16. The dropout rate for MC dropout was kept at $0.2$.

As Fort et al. (2019a) have shown, varying initializations of the weights are most important to getting diverse ensemble members. For this reason, various initialization methods and corresponding parameters were tried, with a Xavier uniform initialization (Glorot & Bengio, 2010) with gain 10, giving the best combination of accuracy and calibration. For more information, refer to Appendix G. This setting is kept for models across all datasets, including the one with an AST backbone.

For the same reason, we investigated whether adding noise to the pre-trained parameters of an Explicit Ensemble increases its performance through a higher diversity of members. However, the results did

not show any additional benefits beyond what the randomly initialized last layer already provided. Therefore, it was not utilized. For more details, refer to Appendix H.

# G  Initialization of LoRA-Ensemble Parameters

Randomness in initialization is a key driver of diversity among ensemble members (Fort et al., 2019a). Therefore, finding the right balance between diversity and overly disrupting parameters is crucial. Hu et al. (2021) propose using a random Gaussian initialization for $A$ while setting $B$ to zero. This approach results in $\Delta W = BA$ being zero at the start of training. In our experiments, we adopt this pattern by always initializing $B$ to zero while varying the parameters and methods for initializing $A$. Following the method outlined by Hu et al. (2021), our initial experiments concentrated on the Gaussian initialization of $A$, with a mean $\mu = 0$ and varying standard deviations. Additionally, we tested the Xavier uniform initialization (Glorot & Bengio, 2010) using different values for the gain. All tests were conducted on the CIFAR-100 dataset and subsequently applied to other experiments. We compared results in terms of accuracy and Expected Calibration Error.

Table 10: Accuracy and Expected Calibration Error for different initialization methods and varying distribution parameters for LoRA-Ensemble.

| Init. Type | Std. / Gain | Accuracy ($\uparrow$) | ECE ($\downarrow$) |
|---|---|---|---|
| Gaussian | 0.02 | 81.2 | 0.041 |
| | 0.05 | 81.4 | 0.037 |
| | 0.1 | 81.7 | 0.035 |
| | 0.2 | 82.1 | 0.034 |
| | 0.5 | 82.6 | 0.036 |
| | 1 | 82.5 | 0.039 |
| | 2 | 81.7 | 0.046 |
| Xavier Uniform | 1 | 81.5 | 0.039 |
| | 5 | 82.2 | 0.034 |
| | 10 | 82.4 | 0.034 |
| | 15 | 82.6 | 0.037 |
| | 20 | 82.4 | 0.038 |
| | 30 | 82.2 | 0.043 |

In Tab. 10, the results are quantitatively presented. It is immediately evident that both techniques and all tested parameters perform similarly. While more specialized models may surpass our results in terms of accuracy, our primary focus is on calibration, with the goal of maintaining comparable predictive performance. Visual inspection of the results in Fig. 13 confirms the high similarity among all results. Choosing a small calibration error while maintaining high accuracy as a decision criterion, both Gaussian initialization with a standard deviation of 0.5 and Xavier uniform initialization with a gain of 10 or 15 are viable candidates. Since a gain of 10 combines high accuracy with the lowest Expected Calibration Error, we select Xavier uniform initialization with a gain of 10 for our experiments.

# H  Initialization of Explicit Ensemble Parameters

A pre-trained Vision Transformer model is the backbone for our computer vision experiments. Correspondingly, the parameters of all members in an Explicit Ensemble are initialized to the same values across members. Initialization is a primary driver of diversity in ensemble members (Fort et al., 2019a). Hence, it is crucial to study the effect of noise in the parameter initialization on the calibration of the resulting ensemble. In the case of pre-trained model weights not having been trained on a dataset with the same number of classes, the last layer of all models is replaced completely. This means that regardless of the ensemble technique used, the weights of the last layer, which is responsible for classification, will vary across members. This variation in the weights of the classification layer is expected to contribute significantly to the diversity of the members. Nonetheless, we studied the impact of adding noise to the parameters of an Explicit Ensemble. This was done

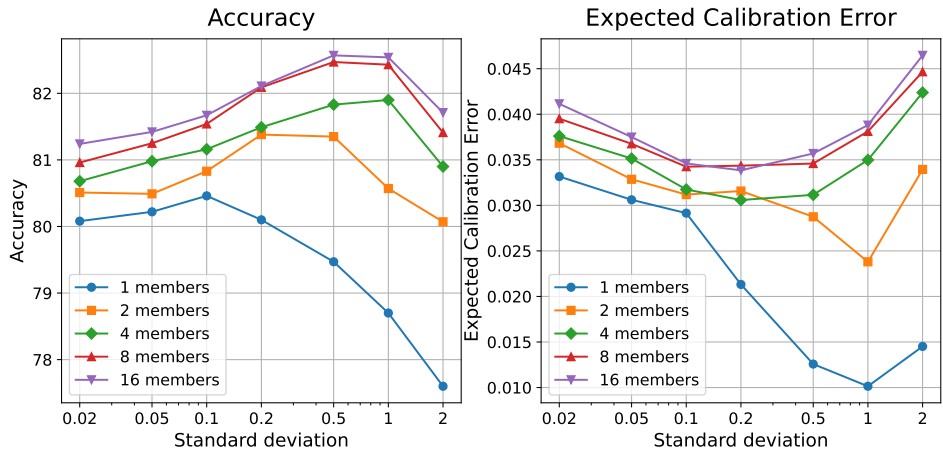

(a) Gaussian initialization with varying standard deviation.

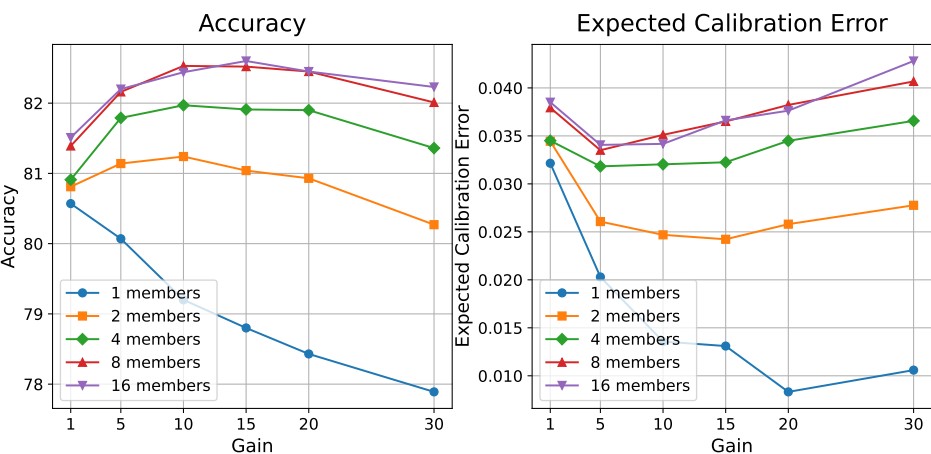

(b) Xavier uniform initialization with varying gain

Figure 13: Accuracy and Expected Calibration Error for different initialization methods and varying distribution parameters across different ensemble sizes for LoRA-Ensemble.

using the following formula:

$$W_{\text{new}} = W + \alpha \cdot dW \,, \tag{6}$$

where $dW \sim \mathcal{N}(0, \sigma_W)$. Here $\alpha$ is a scale factor to control the amount of noise and $\sigma_W$ is the standard deviation of the parameters within a weight matrix. This was applied to all weight matrices separately.

It is expected that the initial layers of a neural network will learn basic features, while the later layers will include dataset-specific properties. Therefore, it is assumed that adding noise to the later layers would increase diversity while maintaining pre-training. However, adding noise to the earlier layers might disrupt pre-training more significantly, especially with smaller datasets, as these parameters may not converge to meaningful values again. To address this, an experiment was set up where noise was added only to the last encoder layers of the model, increasing the number of affected encoder layers gradually. Additionally, several different noise scales $\alpha$ were tried, ranging from 1 to 0.0001. In the presented experiment, the last classification layer is initialized using PyTorch's default method

for linear layers. At the time of writing it is as follows:

$$W_{\text{init}} = \text{Unif}\left(-\sqrt{5} \cdot \sqrt{\frac{3}{fan\_in}}, \sqrt{5} \cdot \sqrt{\frac{3}{fan\_in}}\right) \tag{7}$$

$$B_{\text{init}} = \text{Unif}\left(-\sqrt{\frac{1}{fan\_in}}, \sqrt{\frac{1}{fan\_in}}\right). \tag{8}$$

Here $W$ specifies the weight matrix and $B$ is the bias. Experiments are conducted on the CIFAR-100 dataset.

## H.1 RESULTS

The most important metrics for this section are accuracy and Expected Calibration Error. The results for adding noise to the last layer up to the last five layers are summarized in Fig. 14. Fig. 14a depicts the results for a single model, while Fig. 14b shows the results for an ensemble of 16 members.

It is evident that none of the experiments surpass the baseline of not using any additional noise beyond the random initialization of the last classification layer. After the last five layers, the results become uninteresting, as they do not vary significantly from those shown in the plots. Therefore, the presentation is truncated at five layers. Based on the presented results, no additional noise is injected into the Explicit Ensemble, and only the last layer initialization is varied.

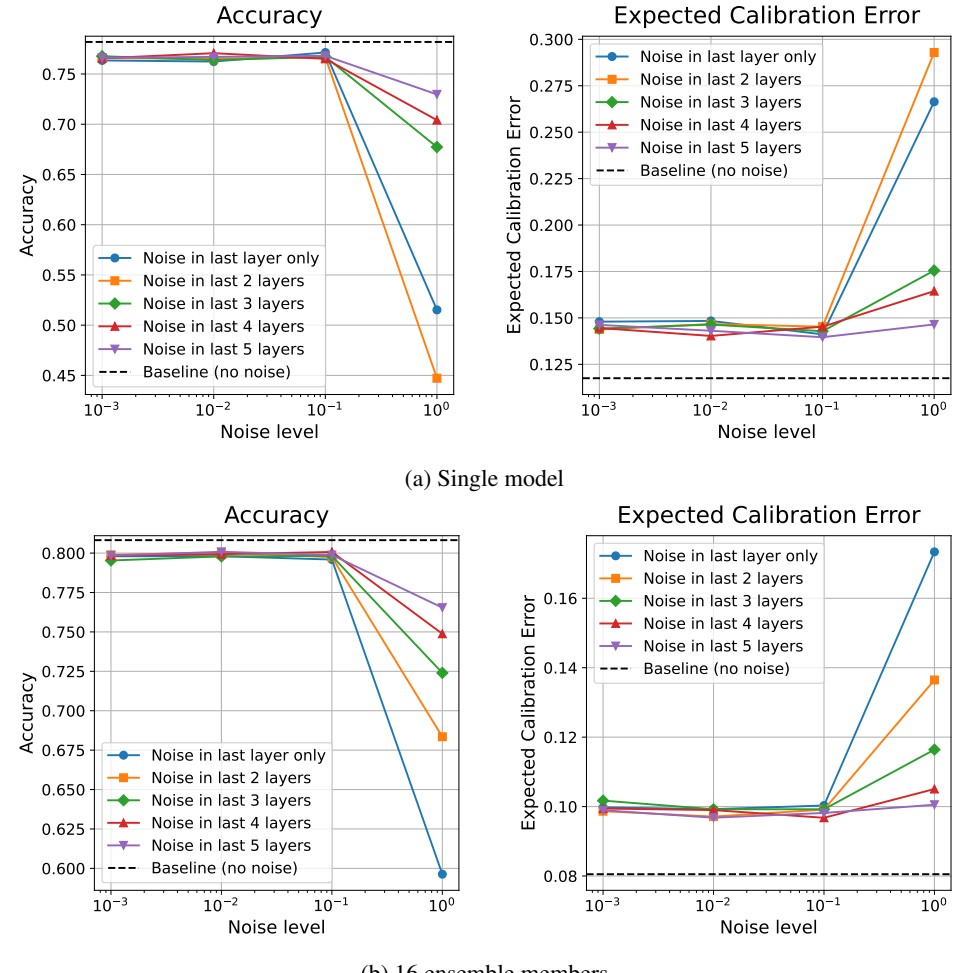

(a) Single model

(b) 16 ensemble members

Figure 14: Accuracy and Expected Calibration Error for different noise levels across varying numbers of layers for the Explicit Ensemble. The baseline with no noise is indicated by a dashed black line.

## I  AST Implementation

A different backbone is used for the experiment on the audio dataset. Specifically, we use the Audio Spectrogram Transformer (AST) following the implementation of Gong et al. (2021), with slight modifications to fit our general architecture. Appendix J demonstrates the equivalence of our implementation. In their experiments, Gong et al. (2021) used two different types of pre-trained weights: one pre-trained on a large image dataset and the other on an audio dataset. For our research, we transfer the weights of a vision transformer model known as DeiT (Touvron et al., 2020), which has been pre-trained on the ImageNet dataset (Deng et al., 2009), to the original AST architecture by Gong et al. (2021). The model has 12 layers, uses 768-dimensional patch embeddings, and the multi-head attention modules have 12 heads. This task is considered more challenging than using models pre-trained on audio datasets.

## J  Validation of AST Implementation

The Audio Spectrogram Transformer (AST) model provided by Gong et al. (2021) was copied without any changes. However, the training and evaluation pipeline was adapted to fit our architecture. Correspondingly, it was essential to validate the equivalence of our implementation by training a single AST on the ESC-50 dataset. The results of our model should closely match those provided in Gong et al. (2021).

They offer two sets of pre-trained weights: one where the weights of a Vision Transformer pre-trained on ImageNet (Deng et al., 2009) are transferred to AST, and another where the AST was pre-trained on AudioSet (Gemmeke et al., 2017). To verify our implementation, we ran it using the settings provided by Gong et al. (2021) and compared the results, which are summarized in Tab. 11. The results for both pre-training modes fall within the uncertainty range provided by Gong et al. (2021). This suggests that our pipeline yields comparable outcomes, validating our implementation for continued use.

Table 11: Comparison of the results obtained for the AST as given in Gong et al. (2021) and those obtained by our implementation. AST-S refers to the AST pre-trained on ImageNet, and AST-P refers to the AudioSet pre-training. Both results fall within the uncertainty range provided by Gong et al. (2021).

| Model | Accuracy (Gong et al., 2021) | Accuracy (our implementation) |
|-------|------------------------------|-------------------------------|
| AST-S | $88.7 \pm 0.7$ | 88.0 |
| AST-P | $95.6 \pm 0.4$ | 95.8 |

## K  Hyper-parameter Tuning for AST Experiment

The original training settings of the AST-S model in Gong et al. (2021) utilize a batch size of 48. However, due to the memory constraint of single GPU training on an NVIDIA Tesla A100 with 80 GB memory, replicating a batch size of 48 as in the original publication was infeasible for training an Explicit AST-S Ensemble with 8 members. Consequently, we perform minimal hyper-parameter tuning by employing a batch size of 1 for both the explicit AST-S and the LoRA AST-S model, exploring various learning rates. Apart from batch size and learning rate adjustments, all other settings remain consistent with Gong et al. (2021).

The hyper-parameter tuning results for the explicit model using a batch size of 1, as shown in Tab. 12, demonstrate performance similar to the original implementation with a batch size of 48, allowing for a fair comparison with our method (Gong et al., 2021). Additionally, Tab. 13 showcases the outcomes of tuning the learning rate for our LoRA AST-S model.

Table 12: Single model 5-Fold cross-validation results of AST-S on ESC-50 sound dataset with different learning rates and batch size 1. The model settings selected based on accuracy for the experiments are **highlighted**.

| Model | Learning rate | Accuracy (↑) | ECE (↓) |
|---|---|---|---|
| **AST-S** | **0.00001** | **88.2** | **0.0553** |
| AST-S | 0.00005 | 81.7 | 0.0933 |

Table 13: Single model 5-Fold cross-validation results for our LoRA AST-S implementation on ESC-50 sound dataset with different learning rates and batch size 1. The model settings selected based on accuracy for the experiments are **highlighted**.

| Model | Learning rate | Accuracy (↑) | ECE (↓) |
|---|---|---|---|
| LoRA AST-S | 0.00001 | 85.6 | 0.0447 |
| **LoRA AST-S** | **0.00005** | **87.9** | **0.0487** |
| LoRA AST-S | 0.0001 | 84.7 | 0.0501 |
| LoRA AST-S | 0.0005 | 24.1 | 0.0291 |
| LoRA AST-S | 0.001 | 11.8 | 0.0295 |

## L COMPUTATIONAL COST FOR AST MODELS

Similarly to the way we did for the Vision Transformer models, we estimate the required resources for AST models. The resource needs are presented in Tab. 14. The number of parameters is reported for an ensemble of 8 members, with the $A$ and $B$ matrices in models using LoRA having a rank of 16. Training and inference times were measured on a single NVIDIA Tesla A100-80GB GPU, with a batch size of 1. Training time is given as the average wall clock time per training epoch while training on ESC-50, with 8 ensemble members. Inference time is reported as the average time for a single forward pass of an ESC-50 sample with a batch size of 1.

As mentioned in Sec. 2.1, the Explicit Ensemble processes the members sequentially, while LoRA-Ensemble is parallelized. However, fully parallelizing the training of AST models causes memory issues, so chunking was introduced. Thus, in LoRA-Ensemble models, the pass through the backbone runs in parallel, while LoRA modules are called sequentially. This also explains the significantly higher inference time compared to the results in Sec. 3.1. Additionally, the one-time delay incurred by PyTorch's *vmap* function causes LoRA-Ensemble to be slightly slower at inference time.

## M HYPERPARAMETER TUNING FOR MC DROPOUT

We conducted an analysis to determine the impact of dropout probability on the accuracy and calibration of the ViT with Monte Carlo dropout. Fig. 15 displays the accuracy and ECE scores for various dropout probabilities. The experiment is carried out on the HAM10000 dataset with 16 members. Our findings show that a dropout probability of 0.2 offers a good balance between accuracy and calibration.

Table 14: Parameter counts and computation times for an Explicit Ensemble of 8 AST models and the corresponding LoRA-Ensemble. Training time is the average duration for one epoch on ESC-50, with batch size 1. Inference time is the average duration of a forward pass, with batch size 1.

| Method | Parameter overhead | Training time [s] | Inference time [ms] |
|---|---|---|---|
| Explicit Ensemble | $8 \times 87M$ | 517 | $8 \times 7.3$ |
| LoRA-Ensemble | $1.08 \times 87M$ | 348 | 73.9 |

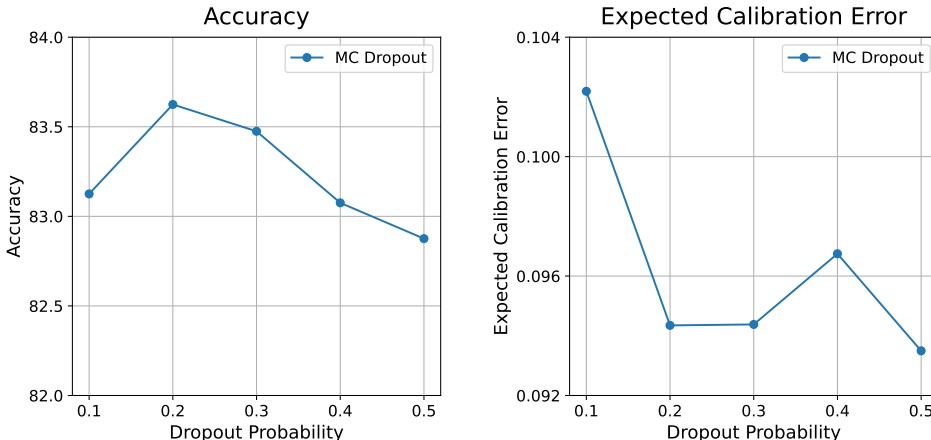

Figure 15: Accuracy and Expected Calibration Error for different dropout probabilities methods for MC Dropout on HAM10000 dataset.

## N  SNAPSHOT ENSEMBLE IMPLEMENTATION DETAILS

Snapshot Ensemble Huang et al. (2017), in its pure form, consists of training a single model with cycling learning and taking snapshots every few epochs. This can make it hard, however, for the model to converge to anything meaningful within the low number of epochs available for training per snapshot. Therefore, Snapshot Ensemble was modified slightly, by first letting training run for a number of epochs, without any cycling of the learning rate. After this burn-in period the learning rate is at 0 and a first snapshot is taken. The remaining number of epochs is split evenly. If the remaining number of epochs is not divisible by the desired number of ensemble members, the burn-in period is extended until it is. For the HAM10000 dataset training is left at 65 epochs, with 20 burn-in epochs. For CIFAR-10 and CIFAR-100 using only 16 epochs would only leave 1 epoch per cycle for bigger models. Therefore, training is extended to 30 epochs with a burn-in period of 15 epochs.

## O  IMPLICIT ENSEMBLE BASELINE CHALLENGE

Many implicit ensemble methods, such as those proposed in Wen et al. (2020); Turkoglu et al. (2022); Durasov et al. (2020); Havasi et al. (2020), are architecture-specific and predominantly designed for MLPs or CNNs. As a result, adapting these techniques to transformer architectures presents significant challenges, since transformers' computation structure is quite different than MLPs and CNNs.

In particular, we attempted to implement FiLM-Ensemble Turkoglu et al. (2022) on a self-attention network, given the promising results reported by its authors. However, the authors themselves noted that applying FiLM-Ensemble to transformers is not straightforward, mainly because transformers rely on LayerNorm, whereas FiLM-Ensemble was developed with BatchNorm in mind. Our experiments confirmed that directly using BatchNorm in transformers led to notable performance degradation. We explored several approaches to adapt LayerNorm, but the most effective results were achieved by fixing all affine parameters for each ensemble member. This allowed for slight initial variations to introduce randomness and diversity, while keeping the variation among members minimal. The results, summarized in Tab. 15, show that increasing the ensemble size slightly improved accuracy, though the Expected Calibration Error (ECE) fluctuated without consistent improvement. In fact, when using larger ensemble sizes, such as 8 or 16, both accuracy and calibration worsened across all settings we tested.

## P  DEFINITIONS OF EVALUATION METRICS

We primarily evaluate our models on accuracy and Expected Calibration Error (ECE, Guo et al., 2017). In addition to accuracy and Expected Calibration Error, we have calculated several other

Table 15: Performance of FiLM-Ensemble for Vision Transformer (ViT) on CIFAR-10. Increasing the ensemble size slightly improves accuracy, but ECE fluctuates without showing consistent improvement.

| # ensemble members | Accuracy ($\uparrow$) | ECE ($\downarrow$) |
|---|---|---|
| 1 | 90.54 | 0.0286 |
| 2 | 91.18 | **0.0269** |
| 4 | **91.23** | 0.0289 |

scores that have been used in the context of probabilistic deep learning. In the following section, we present the formulations used in our implementations.

## P.1   ACCURACY

The accuracy is implemented instance-wise as follows:

$$\text{Acc} = \frac{1}{N} \sum_{i=1}^{N} \frac{|\hat{y}_i \cap y_i|}{|\hat{y}_i \cup y_i|} \tag{9}$$

Here $y_i$ denotes the true label of the sample $i$, $\hat{y}_i$ is the predicted label of the sample $i$, and $N$ means the total number of samples.

## P.2   EXPECTED CALIBRATION ERROR

The Expected Calibration Error is a widely used metric for measuring the calibration of neural networks. We use the definition given in Guo et al. (2017). ECE is defined as the expected difference between accuracy and confidence across several bins. We first need to define accuracy and confidence per bin $B_m$ as follows:

$$\text{Acc}(B_m) = \frac{1}{|B_m|} \sum_{i \in B_m} \mathbf{1}(\hat{y}_i = y_i), \tag{10}$$

$$\text{Conf}(B_m) = \frac{1}{|B_m|} \sum_{i \in B_m} \hat{p}_i. \tag{11}$$

Again, $y_i$ and $\hat{y}_i$ denote the true and predicted labels of sample $i$ respectively, and $\hat{p}_i$ is the predicted confidence of sample $i$. With this the Expected Calibration Error is given as:

$$\text{ECE} = \sum_{m=1}^{M} \frac{|B_m|}{n} |\text{Acc}(B_m) - \text{Conf}(B_m)| \tag{12}$$

## P.3   MACRO F1-SCORE

$$F1 = \frac{1}{C} \sum_{j=1}^{C} \frac{2 p_j r_j}{p_j + r_j}, \tag{13}$$

where $r_j$ represents the Recall of class $j$, defined as $r_j = \frac{TP}{TP+FN}$, and $p_j$ represents the Precision of class $j$, defined as $p_j = \frac{TP}{TP+FP}$, and $C$ refers to the number of classes, Here, $TP$, $FP$, and $FN$ denote True Positives, False Positives, and False Negatives respectively.

## P.4   NEGATIVE LOG-LIKELIHOOD (NLL)

$$NLL = -\frac{1}{N} \sum_{i=1}^{N} \sum_{j=1}^{C} (y_{i,j} \log \hat{p}_{i,j}) = -\frac{1}{N} \sum_{i=1}^{N} \log \hat{p}_i, \tag{14}$$

where $N$ denotes the number of datapoints, $C$ the number of classes, $y_{i,j}$ is 1 if the true label of point $i$ is $j$ and 0 otherwise and $\hat{p}_{i,j}$ is the predicted probability of sample $i$ belonging to class $j$.

## P.5 BRIER SCORE

For Brier score we take the definition by Brier (1950), which is as follows:

$$BS = \frac{1}{N} \sum_{i=1}^{N} \sum_{j=1}^{C} (\hat{p}_{i,j} - y_{i,j})^2, \tag{15}$$

where $N$ denotes the number of datapoints, $C$ the number of classes, $y_{i,j}$ is 1 if the true label of point $i$ is $j$ and zero otherwise and $\hat{p}_{i,j}$ is the predicted probability of sample $i$ belonging to class $j$.

## P.6 AREA UNDER THE RECEIVER OPERATING CHARACTERISTIC CURVE (AUROC)

The AUROC score evaluates the performance of a binary classifier by measuring its ability to distinguish between positive and negative classes, as introduced by Hanley & McNeil (1982). In our out-of-distribution (OOD) detection experiments, the positive class corresponds to an in-distribution sample, while the negative class corresponds to an out-of-distribution sample.

The AUROC is computed as the area under the ROC curve, which plots the true positive rate (TPR) against the false positive rate (FPR) across various decision thresholds. The TPR and FPR are defined as follows:

$$\text{TPR} = \frac{\text{TP}}{\text{TP} + \text{FN}}, \tag{16}$$

$$\text{FPR} = \frac{\text{FP}}{\text{FP} + \text{TN}}, \tag{17}$$

where TP, FP, FN, and TN represent the true positives, false positives, false negatives, and true negatives, respectively.

The AUROC score is given by the following integral:

$$\text{AUROC} = \int_0^1 \text{TPR}(\text{FPR}) \, d\text{FPR}. \tag{18}$$

A higher AUROC score indicates better classification performance, with a score of 1 representing a perfect classifier, and a score of 0.5 indicating performance equivalent to random chance.

## P.7 AREA UNDER THE PRECISION-RECALL CURVE (AUPRC)

The Area Under the Precision-Recall Curve (AUPRC) assesses the performance of a binary classifier by measuring its ability to accurately identify positive instances, as described by Davis & Goadrich (2006). In our out-of-distribution (OOD) detection experiments, the positive class corresponds to in-distribution samples, while the negative class corresponds to out-of-distribution samples.

The AUPRC is calculated as the area under the Precision-Recall (PR) curve, which plots precision against recall at various decision thresholds. Precision and recall are defined as follows:

$$\text{Precision} = \frac{\text{TP}}{\text{TP} + \text{FP}},$$
$$\text{Recall} = \frac{\text{TP}}{\text{TP} + \text{FN}},$$

where TP, FP, and FN represent true positives, false positives, and false negatives, respectively.

The AUPRC score is the integral of precision with respect to recall, expressed as:

$$\text{AUPRC} = \int_0^1 \text{Precision(Recall)} \, d\text{Recall}.$$

A higher AUPRC score indicates better classifier performance in recognizing positive instances, with a score near 1 representing a good classifier, characterized by both high recall and high precision. This metric is especially valuable for evaluating classifiers on imbalanced datasets.

