# OpenReview forum: "LoRA-Ensemble: Efficient Uncertainty Modelling for Self-attention Networks"
_ICLR.cc/2025/Conference — Submitted to ICLR 2025_

### Official Review · Reviewer_tSvT · 2024-10-31

**Soundness:** 3
**Presentation:** 3
**Contribution:** 2
**Rating:** 5
**Confidence:** 4

**Summary:**

In this paper, the authors propose a novel method for uncertainty estimation in neural networks based on the low-rank adaptation (LoRA) approach. In a nutshell, the method trains several sets of low-rank matrices (commonly used for adaptation/fine-tuning of a pretrained model) and, during inference, runs predictions using all of them, then averages the outputs. Experiments on several benchmarks across various domains demonstrate that the proposed method outperforms several popular ensembling approaches, including (in some cases) deep ensembling. The method improves the final model's performance (e.g., in terms of accuracy), aleatoric uncertainty (measured by calibration metrics), and epistemic uncertainty (in terms of OOD detection quality). At the same time, the method induces significantly lower computational and memory overhead, as well as faster training, compared to conventional ensembles. The proposed approach is relatively simple to use with various attention-based architectures, although it is somewhat limited to them.

**Strengths:**

* The paper is clearly written and easy to follow. The idea is intuitive and easy to grasp. The related work section provides an adequate discussion of existing approaches to both LoRA and uncertainty estimation for deep learning. The analysis narrative, with the presented drawbacks of existing methods (such deep ensembles), is very clear and easy to understand

* The method is straightforward to apply, requiring only a (potentially pretrained) model and the training of a number of LoRA adapters. During inference, predictions from these adapters are averaged, making implementation efficient and compatible with existing architectures

* The approach is significantly more efficient than traditional (deep-)ensembling, offering reduced computational and memory overhead while maintaining strong performance

* The method demonstrates strong results across a wide range of benchmarks, showing its versatility. This includes performance on image classification tasks (such as CIFAR and HAM10000), as well as on audio classification (ESC-50).

**Weaknesses:**

* The method is primarily limited to attention-based models, restricting its applicability across model architectures.

* A potential limitation is the method's similarity to existing method BatchEnsembles, as both approaches utilize low-rank adapters to create ensembles. Given this mentioned similarity a question arises: what advantages does the proposed method offer over directly applying BatchEnsembles to attention heads?

* Experiments demonstrates mixed results, making it unclear whether the method consistently outperforms existing approaches. For example, in CIFAR experiments it’s unclear where the significant performance improvement over a single model (without a LoRA adapter) comes from. The calibration performance in terms of ECE may also stem from discrepancies in accuracy. Similarly, ESC-50 experiments show that deep ensembles outperform the proposed method.

* In addition to the previous point, the authors do not compare against other compute-efficient ensembling approaches. In the appendix, they mention that existing methods cannot be applied to Transformer architectures, as these differ significantly from MLP and CNN-based models. However, BatchEnsembles or Masksembles (and not mentioned PackedEnsembles [1]) could be viewed as approaches that apply perturbations to trainable model weights (matrices), therefore the impossibility to apply them to Transformers is unclear.

* An important limitation is the absence of experiments on text data, despite LoRA’s popularity and proven effectiveness within this domain.

[1] Laurent, Olivier, et al. "Packed-ensembles for efficient uncertainty estimation." ICLR (2023).

**Questions:**

* How effective is using a mixture-of-experts approach as an “ensemble” for uncertainty evaluation, and how does it compare to this method in terms of efficiency and uncertainty quality?

* Given the method’s focus on attention-based models, how well does it work for MLPs or CNNs?

* What specific advantages does this method offer over applying BatchEnsembles to attention heads?

* The appendix suggests existing efficient ensembling methods don’t apply to Transformers. What specifically prevents methods like BatchEnsembles, Masksembles, or PackedEnsembles from being effective here?

---

> ### Author Response · Authors · 2024-11-27
> **Rebuttal, part 1**
>
> We thank the reviewer for the detailed and constructive feedback. Below, we address all the concerns in detail.
>
> **W1. The method is primarily limited to attention-based models…**
>
> Developing a generic algorithm that works seamlessly across all types of network architectures, such as RNNs, CNNs, MLPs, and Transformers, is extremely challenging when the algorithm involves modifying the structure of computation. To the best of our knowledge, no existing work demonstrates such applicability across all architectures for methods that fundamentally alter network architecture.
>
> Our work explicitly focuses on self-attention networks, as reflected in the title: LoRA-Ensemble: Efficient Uncertainty Modelling for Self-Attention Networks. This focus was intentional, given the widespread adoption and popularity of attention-based models in a variety of domains. By constraining ourselves to this architecture, we aim to provide a robust and efficient solution tailored to its specific structure and use cases.
>
> Additionally, while our method is limited to self-attention models, we demonstrate its versatility across multiple domains, including medical image classification and sound classification. Furthermore, we evaluate our approach on different versions of transformers (e.g., ViT, DeiT) and compare their performance to validate its robustness and flexibility. These experiments highlight that our method can address a wide range of problems across diverse application areas, showcasing its broad applicability within the realm of attention-based models.
>
>
> **W2. & Q3 Similarity to existing method BatchEnsembles… - What specific advantages..**
>
> While there is a technical similarity, as noted in the Related Work section, where both methods constrain the parameter space using a low-rank approach, the methods are fundamentally different in design. First, the formulations are distinct, which already makes the two approaches quite different. In deep learning, formulation is critical, similar to how GRU and LSTM are considered separate architectures despite slight differences in their computational structure.
>
> Furthermore, the two methods are developed for entirely different architectures. BatchEnsemble was designed for MLPs, whereas our method is specifically developed for Transformers. This architectural difference further highlights the distinct nature of the methods. Additionally, there are numerous technical distinctions, such as how the parameters are initialized and how pre-trained weights are utilized in our method, which is not the case for BatchEnsemble. Beyond introducing a new ensembling method, our work also explains why it performs well, as detailed in the newly added Section 4. Considering these differences in formulation, experimental focus, and analysis, we believe there is no concern regarding the novelty of our paper.
>
> While re-implementing the BatchEnsemble idea for comparison would be interesting, adapting this algorithm to work effectively with Transformers would require significant re-engineering efforts, careful design choices, and substantial experimentation, which is beyond the scope of this work.
>
> **W3.1. It is unclear where the performance gain comes from…**
>
> To provide better insight into the effectiveness of LoRA-Ensemble, we have added a new section in the Appendix (see Section A.4, Table 7), where we compare a high-rank LoRA model with a low-rank LoRA model that have a similar number of trainable parameters. The results clearly demonstrate that the low-rank LoRA model outperforms its high-rank counterpart, showing that the performance improvement is not solely attributable to the number of trainable parameters. Instead, the results highlight the unique learning dynamics enabled by LoRA-Ensemble.
>
> Additionally, we have introduced a new section, "Enhanced Diversity in LoRA-Ensemble," which empirically analyzes why LoRA-Ensemble outperforms methods like Explicit Ensemble. This section provides a much deeper understanding of the underlying reasons behind LoRA-Ensemble’s superior performance across most of the experimental settings. Please refer to Section 4 for further details.
>
>
> **W3.2. Mixed results… - ESC-50 experiments**
>
> We would like to clarify that, while we do not claim that LoRA-Ensemble outperforms all the baselines in every setting, we do demonstrate that LoRA-Ensemble outperforms or is on par with Explicit Ensemble, while being significantly more computationally efficient. In the ECS-50 experiment, although Explicit Ensemble achieves better performance on most metrics, the differences between LoRA-Ensemble and Explicit Ensemble are statistically insignificant.
>
> We also provide a potential explanation for this in the paper: the initial pre-training was conducted on ImageNet, where there is a large distributional shift between pre-trained dataset and target dataset. We believe if the models were initialized with more appropriate pre-training, LoRA-Ensemble would outperform for this task as well.

---

> ### Author Response · Authors · 2024-11-27
> **Rebuttal, part 2**
>
> **W4. & Q4. Existing methods cannot be applied to Transformer architecture**
>
> We would like to clarify that we do not claim it is impossible to adapt methods like BatchEnsemble, or similar approaches to transformers. Instead, as mentioned in Appendix M, we state that it is challenging due to the unique computational structure of transformers compared to CNNs and MLPs. To illustrate, we attempted to implement FiLM-Ensemble on transformers, which required significant time and effort. While we achieved some preliminary results by adapting LayerNorm to emulate FiLM-Ensemble's behavior, the improvements were minimal and inconsistent.
>
> Even for simpler methods like Dropout, adapting them to transformers is not straightforward. For example, significant efforts have been made to adapt Dropout to transformers, such as DropAttention (Lin et al., 2019) and DropKey (Li et al., 2023). These approaches involve substantial modifications to the standard Dropout technique, including mechanisms like re-normalization in DropAttention and advanced dropping schedules in DropKey, to make them compatible with transformers. This highlights that even the simplest implicit ensembling methods often require significant re-engineering for transformers.
>
> Furthermore, methods like BatchEnsemble have not shown promising results, even on simpler architectures like ResNets. For example, on CIFAR-10, BatchEnsemble achieves a performance of 95.9 compared to 95.7 for MC-Dropout and 95.3 for a single network. While slightly better than MC-Dropout, the improvement is quite insignificant, with no reported error bars, and the Expected Calibration Error (ECE) does not show meaningful improvement either. Additionally, these methods often rely on limited experimental evaluations and do not establish themselves as robust or generalizable approaches.
>
> Given these challenges and the lack of promising results from these methods, we do not see the merit in re-implementing them for transformers, which is itself an active area of research, when their existing performance does not justify such effort. Additionally, we have now included PackedEnsembles in the Related Work section for completeness.
>
> **References**
>
> Lin, Zehui, Pengfei Liu, Luyao Huang, Junkun Chen, Xipeng Qiu, and Xuanjing Huang. "DropAttention: A regularization method for fully-connected self-attention networks." arXiv preprint arXiv:1907.11065, 2019.
>
> Li, Bonan, et al. "DropKey for vision transformer." Proceedings of the IEEE/CVF Conference on Computer Vision and Pattern Recognition, 2023.
>
>
> **W5. Absence of experiments on text data**
>
> Thank you for highlighting the absence of experiments on text data. While LoRA has indeed been popular in the text domain, our work primarily focuses on developing an ensemble approach applicable across diverse tasks, rather than targeting any specific domain like natural language processing. To evaluate uncertainty and demonstrate the robustness of our method, we chose well-established benchmarks such as CIFAR-10 and CIFAR-100, a real-world medical image dataset, and out-of-distribution (OOD) tasks. These datasets and tasks are widely used in the literature on ensemble methods, ensuring that our evaluation is in line with existing works and provides a practical and comprehensive test bed for assessing the generalizability of our approach.
> Incorporating text data is certainly an interesting direction, but integrating additional domains such as NLP would require significant extensions beyond the scope of this paper. We aim to ensure our experiments remain focused and aligned with the primary goal of developing an efficient ensembling method for self-attention networks. That said, we appreciate the suggestion and may explore applications in the text domain as part of future work.
>
>
> **Q1. How effective is using a mixture-of-experts approach as an “ensemble” for uncertainty evaluation**
>
> Comparing our method to mixture-of-experts approaches in terms of efficiency and uncertainty quality would be an intriguing direction for future work. However, such comparisons are beyond the scope of this paper. Expanding the discussion to include mixture-of-experts and other uncertainty estimation techniques, such as Bayesian approaches, would significantly broaden the focus and dilute the paper’s emphasis on efficient ensembles for self-attention networks. [1] can be an interesting read for you.
>
> **References**
>
> [1] Jiang, Yuchang, et al. "Mixture of Experts with Uncertainty Voting for Imbalanced Deep Regression Problems." arXiv preprint arXiv:2305.15178 (2023).
>
> **Q2. how well does it work for MLPs or CNNs?**
>
> It would indeed be interesting to explore how the method performs for MLPs or CNNs. However, this is beyond the scope of our paper, as our work specifically targets efficient ensembles for self-attention networks, as reflected in the title: LoRA-Ensemble: Efficient Uncertainty Modelling for Self-Attention Networks. This could be an interesting direction for future work.

---

> > ### Comment · Reviewer_tSvT · 2024-12-02
> >
> > First of all, I would like to thank the authors for providing a detailed rebuttal. Here are some comments regarding it:
> >
> > * In the rebuttal, the authors mentioned that "Developing a generic algorithm that works seamlessly across all types of network architectures, such as RNNs, CNNs, MLPs, and Transformers, is extremely challenging." However, this is precisely the goal of all efficient ensembling methods. For example, the BatchEnsemble approach is highly versatile, and I cannot think of an architecture to which it could not be applied (including Transformers).
> >
> > * The claim that "BatchEnsemble was designed for MLPs, whereas our method is specifically developed for Transformers" is inaccurate. The BatchEnsemble approach can be adapted to convolutional operations, and Transformer blocks, in fact, contain linear layers to which BatchEnsemble can be directly applied (in the same way LoRA is). Unless I am missing something, the only difference between LoRA-Ensemble and BatchEnsemble lies in how they compute perturbations for the original weights: the former uses additive perturbations, while the latter uses multiplicative ones. While this difference is worth acknowledging, if it is the primary distinction, it significantly limits the novelty of the paper.
> >
> > * Considering the above, adding BatchEnsemble as one of the baselines seems necessary. Otherwise, the absence of this comparison, combined with the novelty concerns, weakens the overall contribution.
> >
> > I value the authors' contribution, and it is interesting, but given the discussed issues, I am not confident about increasing the score.

---

### Official Review · Reviewer_vyv1 · 2024-11-03

**Soundness:** 3
**Presentation:** 3
**Contribution:** 2
**Rating:** 5
**Confidence:** 2

**Summary:**

The paper introduces LoRA-Ensemble, an method for parameter-efficient ensebmling for self-attention networks. Key idea is to have different low-rank matrices for each ensemble component keeping the backbone frozen. The LoRA matrices are applied only to the self-attention layer making the proposed approach parameter efficient. The proposed method is computationally efficient, improves over traditional ensemble methods, and improves both the accuracy and calibration for multiple classification tasks.

**Strengths:**

- The paper is generally well written and easy to follow.
- The proposed ensemble method: LoRA-Ensemble is intuitive, and simple. It can easily be extended to different transformer-based classification tasks.
- LoRA-Ensemble improves both the predictive accuracy and the calibration performance on the considered classification tasks.
- Propoed method is empirically validated and is parameter/computation efficient.

**Weaknesses:**

- Limited Theoretical insights and empirical validation: The proposed approach is heuristic and is only empirically validated on limited datasets that are relatively simple. Though challenging, the work could be strengthened with some theoretical analysis and guarantees for the proposed technique.  Alternatively, the work could be significantly strengthened by carrying out experiments on more challenging vision datasets of Fine-Grained Visual Classification tasks (see Visual Prompt tuning by Jia et al), Visual Task Adaptation Benchmark/VTAB ( A large-scale study of representation learning with the visual task adaptation benchmark, Zhai et al ), and/or tasks beyond classification (maybe for segmentation/object detection).

- Paper formatting and presentation could be improved: For eg. Table 2, Table 3, Table 7 all overflow.

- For codes, it would have been helpful if codes were included as supplementary material, or at an annonymized github link.

- Post-hoc and other calibration techniques be introduced to further improve the calibration/under-confidence issue, and miscalibration in the baseline models. Discussion and empirical analysis of calibration techniques seems to be missing.

- Some important works in  epistemic uncertainty estimation are missing: The authors miss literature on Evidential Deep Learning (A Comprehensive Survey on Evidential Deep Learning and Its Applications), and second order based UQ  (Second-Order Uncertainty Quantification: A Distance-Based Approach). These works propose computationally efficient alternatives for UQ, can be extended to DL approaches, and can enable UQ a single forward pass as they do not involve the computational overhead of ensemble. Discussion, and comparison with these works could further strengthen the work.

**Questions:**

Please see and address the comments on the weakness section. Additionally, here are some clarifying questions I have.
- Considering the relatively simple datasets, how would the approach scale to large datasets such as imagenet?
- For OOD experiment with Cifar100, it would be interesting to see performance on SVHN as the OOD dataset. I think SVHN is more realistic OOD for Cifar100/Cifar10 comapred to using CIfar10 as OOD for Cifar100.

---

> ### Comment · Reviewer_vyv1 · 2024-11-27
> **Final Score**
>
> No rebuttal was provided by the authors and my concerns remain.
> I've decided to lower the score to 4 (weak reject).

---

> ### Author Response · Authors · 2024-11-27
> **Rebuttal**
>
> We apologize for any confusion and would like to clarify that there may have been a misunderstanding regarding the rebuttal deadline. ICLR extended the discussion period by an additional 6 days, until December 3rd. As a result, we did not submit our rebuttal by the original deadline, as we were focused on completing the revision of the paper. We kindly request that our response be considered for a fair evaluation of our work, given the extended rebuttal period.
>
>
> We sincerely thank the reviewer for their thoughtful feedback and valuable suggestions, below you can find our response to individual concerns in detail.
>
>
> **W1.1. Limited Theoretical insights and empirical validation**
>
> Providing a theoretical analysis of LoRA-Ensemble's relationship with the underlying probability distribution and its ability to prevent ensemble member degeneration is highly challenging. This difficulty arises from the complex interactions in high-dimensional parameter spaces, where pre-trained weights, LoRA updates, random initialization, optimization dynamics, and non-linear transformations collectively shape the learning process. Developing an analytical model would require strong assumptions about data, optimization trajectories, and architecture, which may not universally apply and would constitute a significant contribution to the field in itself.
>
> Since such an analytical proof is beyond the scope of this work, we have instead added a new section, "Enhanced Diversity in LoRA-Ensemble," which empirically analyzes why LoRA-Ensemble outperforms methods like Explicit Ensemble. This section investigates the diversity of ensemble members in both function and weight spaces. The results demonstrate that LoRA-Ensemble achieves significantly greater diversity compared to Explicit Ensemble, which is crucial for constructing effective ensemble.
>
> Additionally, our spectral analysis of the weight space reveals that LoRA-Ensemble produces distinct parameter updates by exploring new high-ranking dimensions that are near orthogonal to the initial weights. In contrast, Explicit Ensemble members tend to adhere more closely to the structure of the initial weights. This enhanced diversity enables LoRA-Ensemble to explore the loss landscape more effectively and capture a broader range of representations, leading to improved robustness and epistemic uncertainty estimation. We also provide visualizations of the training evolution of ensemble members in the loss landscape, demonstrating that LoRA-Ensemble members converge across a broader area, reflecting diverse learning dynamics.
>
> **W1.2 & Q1. Large Scale Datasets**
>
> We are currently conducting large-scale experiments on the iNaturalist dataset to demonstrate the scalability and robustness of our method on large, unbalanced, and challenging datasets. Due to computational constraints, the results are not yet complete, but our initial findings are promising, showing improvements over Explicit Ensemble baselines. We commit to including the full results for iNaturalist in the final version of the paper.
>
>
> **W2. Paper Formatting**
>
> We fixed formatting.
>
> **W3. Code**
>
> The PyTorch implementation of LoRA-Ensemble, along with a detailed README file, has already been included in the supplementary zip file.
>
>
> **W4. Post-hoc techniques**
>
> We have added a new section, Appendix E, to address this concern. We conducted analysis of the impact of Temperature Scaling (TS) as a post-hoc method for improving the calibration of the evaluated approaches. The results indicate that TS enhances calibration across all compared methods, except for the Single Net w/ LoRA.
> Importantly, while post-hoc methods like temperature scaling further improve calibration, the LoRA-Ensemble consistently outperforms all other compared methods, maintaining its superior performance in both diversity and uncertainty estimation.
>
>
>
> **W5. Missing Literature**
>
> Thank you for your suggestion. While we appreciate the importance of works like EDL and second-order UQ, incorporating them would be beyond the scope of this paper, as our focus is specifically on ensemble-based approaches. Including such methods would significantly broaden the scope, potentially leading to a loss of focus, as other uncertainty quantification techniques, such as Bayesian approaches, would also need to be considered.
>
>
> **Q2. OOD experiment**
>
> To address this concern, we added a new experiment. Please see Table 5 for results. We conducted an additional OOD experiment using CIFAR-100 as the in-distribution dataset and SVHN as the out-of-distribution dataset. Our method significantly outperforms all other methods. Notably, LoRA-Ensemble achieves approximately a 10% increase in both metrics compared to the closest baseline. In this experiment, the performance of Explicit Ensemble degrades compared to a single network. This finding aligns with the results reported for Split-Ensemble, where Explicit Ensemble performance also degrades under the same OOD scenario.

---

> > ### Comment · Reviewer_vyv1 · 2024-12-02
> > **Update on the Rebuttal**
> >
> > I thank the authors for the clarification regarding the extended deadline, and providing the rebuttal. Some of my concerns have been addressed. However, some concerns remain.
> > For instance, experimental results in more realistic and challenging datasets [More practical benchmarks could be Fine-Grained Visual Classification tasks (see Visual Prompt tuning by Jia et al), Visual Task Adaptation Benchmark/VTAB ( A large-scale study of representation learning with the visual task adaptation benchmark, Zhai et al ), ImageNet] can strongly strengthen the work. Without looking at the empirical results, it is challenging to accurately access the gain from the proposed methodology.
> > I keep my original rating of 5 [marginally below the acceptance threshold].

---

### Official Review · Reviewer_tWzW · 2024-11-04

**Soundness:** 2
**Presentation:** 3
**Contribution:** 2
**Rating:** 5
**Confidence:** 3

**Summary:**

In this paper, the authors address the challenge of overconfident and uncalibrated predictions in machine learning algorithms, specifically in state-of-the-art neural networks like transformers. They propose a parameter-efficient deep ensemble method called LoRA-Ensemble, which is based on Low-Rank Adaptation (LoRA). This method emulates the ensemble model without the need for separate ensemble members, making it computationally and memory efficient. By training member-specific low-rank matrices for attention projections in a single pre-trained self-attention network, LoRA-Ensemble achieves superior calibration compared to explicit ensembles and performs well across various prediction tasks and datasets. The authors demonstrate the effectiveness of LoRA-Ensemble in different classification tasks, including image labeling, skin lesion classification, sound classification, and out-of-distribution detection.

**Strengths:**

1.	Improved uncertainty calibration: the authors demonstrate that LoRA-Ensemble achieves superior calibration compared to explicit ensembles. This means that the predicted probabilities of the model align more closely with the true probabilities, resulting in more reliable and accurate predictions.
2.	Computational and memory efficiency: One strength of the LoRA-Ensemble method is its ability to emulate an ensemble model without the need for separate ensemble members. This makes it computationally and memory efficient compared to explicit ensemble methods, which require training and storing multiple models.

**Weaknesses:**

1.	Theoretical analysis is lacking regarding the relationship between the LoRA ensemble and probability distribution. It would be beneficial to provide a theoretical proof of how the LoRA ensemble can approximate the underlying distribution more effectively.
2.	Additionally, the analysis on why the LoRA ensemble can prevent the degeneration of ensemble members into a point estimate is missing. It would be helpful to include a discussion on how the diversities of ensemble members are maintained through random initialization.
3.	Lack of evaluation on large-scale datasets. LoRA may face overfitting issues, especially when applied to complex tasks. It would be valuable to assess its performance on larger datasets, such as ImageNet-1K, to gain a better understanding of its capabilities in such scenarios.

**Questions:**

While it is reasonable to use CIFAR100 as the in-distribution data and CIFAR-10 as the out-of-distribution data, it would be more comprehensive to include additional out-of-distribution datasets as well. Additionally, it would be beneficial to provide a detailed comparison with existing benchmarks, such as OpenOOD[1], which is widely used for evaluating out-of-distribution detection methods.
[1] OpenOOD: Benchmarking Generalized Out-of-Distribution Detection. arXiv:2210.07242 [cs.CV]

---

> ### Author Response · Authors · 2024-11-27
> **Rebuttal**
>
> We thank the reviewer for the detailed and constructive feedback. Below, we address all the concerns in detail.
>
> **W1 & W2. Theoretical Analysis and Degeneration of Ensemble Members**
>
> Providing a theoretical analysis of LoRA-Ensemble's relationship with the underlying probability distribution and its ability to prevent ensemble member degeneration is highly challenging. This difficulty arises from the complex interactions in high-dimensional parameter spaces, where pre-trained weights, LoRA updates, random initialization, optimization dynamics, and non-linear transformations collectively shape the learning process. Developing an analytical model would require strong assumptions about data, optimization trajectories, and architecture, which may not universally apply and would constitute a significant contribution to the field in itself.
>
> Since such an analytical proof is beyond the scope of this work, we have instead added a new section, "Enhanced Diversity in LoRA-Ensemble," which empirically analyzes why LoRA-Ensemble outperforms methods like Explicit Ensemble. This section investigates the diversity of ensemble members in both function and weight spaces, using metrics such as pairwise disagreement rates, Jensen-Shannon divergence, and weight-space cosine similarity. The results demonstrate that LoRA-Ensemble achieves significantly greater diversity compared to Explicit Ensemble, which is crucial for constructing an effective ensemble.
>
> Additionally, our spectral analysis of the weight space reveals that LoRA-Ensemble produces distinct parameter updates by exploring new high-ranking dimensions that are near orthogonal to the initial weights. In contrast, Explicit Ensemble members tend to adhere more closely to the structure of the initial weights. This enhanced diversity prevents LoRA-Ensemble from converging to a point estimate and enables it to explore the loss landscape more effectively, capturing a broader range of representations and leading to improved robustness and epistemic uncertainty estimation. We also provide visualizations of the training evolution of ensemble members in the loss landscape, demonstrating that LoRA-Ensemble members converge across a broader area, reflecting diverse learning dynamics.
>
> **W3. Large Scale Datasets.**
>
> To further strengthen our evaluation, we are currently conducting large-scale experiments on the iNaturalist dataset to demonstrate the scalability and robustness of our method on large, unbalanced, and challenging datasets. Due to computational constraints, the results are not yet complete, but our initial findings are promising, showing improvements over Explicit Ensemble baselines. We commit to including the full results for iNaturalist in the final version of the paper.
>
>
>
> **Q1.OOD Dataset.**
>
> To address this concern, we added a new experiment. Please see Table 5 for the results. We conducted an additional out-of-distribution (OOD) experiment using CIFAR-100 as the in-distribution dataset and SVHN as the out-of-distribution dataset, as this is a well-established benchmark in the literature. The results further emphasize the potential of LoRA-Ensemble for OOD tasks, as it significantly outperforms all other compared methods, including Explicit Ensembles and Split-Ensembles, in terms of AUROC and AUPRC. Notably, LoRA-Ensemble achieves approximately a 10% increase in both metrics compared to the closest baseline, Split-Ensemble. In this experiment, the performance of Explicit Ensemble degrades compared to a single network. This finding aligns with the results reported for Split-Ensemble, where the performance of Explicit Ensemble also degrades under the same OOD scenario. These results highlight the robustness of LoRA-Ensemble and demonstrate its superiority for OOD tasks.

---

> > ### Comment · Reviewer_tWzW · 2024-12-02
> >
> > Although the authors tried their best to supplement the experiments and provide explanations, some of the experimental results have not yet been completed. I will maintain my previous score.

---

### Official Review · Reviewer_Wiq6 · 2024-11-04

**Soundness:** 3
**Presentation:** 3
**Contribution:** 1
**Rating:** 3
**Confidence:** 4

**Summary:**

The paper creates ensembles by starting from a base model and adapting each member independently using LoRA adapters. Each LoRA module adds a small, low-rank learnable matrix to the MLP of Transformers. Experiments on datasets like CIFAR-100, HAM10000, ESC-50 show that these ensembles can give better performance and calibration.

**Strengths:**

* S1. Creating lightweight ensembles, requiring fewer resources at training time and test time is a good and relevant direction.

* S2. The approach of using LoRA in ensembles is sound and has good potential.

**Weaknesses:**

* W1. The approach is simple, ensembling is a classic technique and LoRA is one of the most used ways to fine-tune a pre-trained model in the context of large models. Having a simple method is usually a good aspect, but when the method is straightforward we need a higher bar for results and investigations. In this paper, experiments are mostly made on small datasets and it doesn’t seem like we gain that much from them. Also, combining ensembling and LoRA has been investigated in the context of LLMs [A]. It doesn’t seem like this paper has any additional insights. The authors can also check the reviews of [A] on Openreview for additional related work.

* W2. Small scale experiments. The experiments in this paper are rather on a small scale. Given the simplicity of the method, we need to evaluate it more thoroughly.

* W3. More baselines are needed. It seems like adding a LoRA module to Single Network gives an important boost, comparable to the ensembling (3% vs 5.9% on Cifar100 classification, 4.8% vs 6.5% on AURPC in OOD detection). It seems that benefits come from adding more learnable parameters, either by LoRA or by ensembling. Thus a single Single Network with a bigger LoRA module (similar number of parameters as the LoRA ensemble) would be a good baseline. Training single, or ensemble method with the same number of parameters would also represent good additional baselines.

[A] Wang et al. LoRA ensembles for large language model fine-tuning. 2023
https://openreview.net/forum?id=X5VElAKt2s

**Questions:**

Did the authors did any comparisons where they fix the number of parameters, or computational budget (at training or testing) between LoRA ensembles and other baselines? Such comparisons would be very relevant.

---

> ### Comment · Reviewer_Wiq6 · 2024-11-27
> **Final comment**
>
> Since there is no rebuttal and the reviewers are in consensus I maintain my initial assessment and score.

---

> ### Author Response · Authors · 2024-11-27
> **Rebuttal**
>
> We apologize for any confusion and would like to clarify that there may have been a misunderstanding regarding the rebuttal deadline. ICLR extended the discussion period by an additional 6 days. As a result, we did not submit our rebuttal by the original deadline, as we were focused on completing the revision of the paper. We kindly request that our response be considered for a fair evaluation of our work, given the extended rebuttal period.
>
>
> We thank the reviewer for the detailed and constructive feedback. Below, we address all the concerns in detail.
>
> **W1.1 Additional Insights**
>
> We significantly extend our analysis in the newly added section, "Enhanced Diversity in LoRA-Ensemble," where we investigate the diversity of ensemble members in both function and weight spaces. Using metrics such as pairwise disagreement rates, Jensen-Shannon divergence, and weight-space cosine similarity, we show that LoRA-Ensemble achieves significantly greater diversity compared to Explicit Ensemble, which is crucial for constructing an effective ensemble.
> In light of our spectral analysis of the weight space, the results reveal that LoRA-Ensemble leads to very distinct parameter updates compared to Explicit Ensemble. LoRA-Ensemble explores new high-ranking dimensions that are near orthogonal to the initial weights, while Explicit Ensemble tends to adhere more closely to the structure of the initial weights. This enhanced diversity allows LoRA-Ensemble to better explore the loss landscape and capture a broader range of representations, leading to improved robustness and epistemic uncertainty estimation. Additionally, we provide visualizations of the training evolution of ensemble members in the loss landscape, demonstrating that LoRA-Ensemble members converge across a broader area, reflecting diverse learning dynamics.
>
>
> **W1.2 Combining ensembling and LoRA**
>
> Science often progresses through multiple groups independently exploring similar ideas, which highlights the significance and relevance of this research direction. Wang et al. work is a concurrent preprint that has not undergone peer review and focuses on fine-tuning LLMs on small text corpora with limited experimental evaluation and comparison. Additionally, their implementation details are unclear, and no code is available, making it difficult to fully assess their approach. In contrast, our work provides a much broader scope and deeper analysis. We conduct experiments across five datasets with diverse application domains, including real-world tasks where epistemic uncertainty is crucial, such as medical image analysis. We also evaluate our method on out-of-distribution scenarios, demonstrating its robustness and generalization. Finally, our work also investigates key factors such as initialization strategies and LoRA rank, which play a crucial role in the method’s effectiveness. By presenting a more extensive evaluation, covering diverse application domains, and introducing insights into the method’s design, our study significantly advances the understanding and applicability of LoRA-Ensemble.
>
> **W2. Small scale experiment**.
>
> To further strengthen our evaluation, we are currently conducting large-scale experiments on the iNaturalist dataset to demonstrate the scalability and robustness of our method on large, unbalanced, and challenging datasets. Due to computational constraints, the results are not yet complete, but our initial findings are promising, showing improvements over Explicit Ensemble baselines. We commit to including the full results for iNaturalist in the final version of the paper.
>
> **W3. Q1. More baselines**
>
> Thank you for this feedback. In response, we have added a new section in the Appendix (see Sec. A.4, Table 7) comparing a high-rank LoRA model with a low-rank LoRA model that have a similar number of trainable parameters. The results demonstrate that the low-rank LoRA model outperforms the high-rank counterpart, highlighting that the performance improvement is not solely due to the number of trainable parameters.
> This finding is in line with our new analysis and the analysis by Shuttleworth et al. (2024) [1], which show that as the rank of LoRA increases, its parameter update structure begins to resemble that of explicit ensemble models (or single model if #member=1), closely adhering to the structure of the initial weights. This similarity reduces the effectiveness of LoRA's unique low-rank adaptation capabilities. Additionally, we provide a new insight about this phenomenon in the Discussion section, further emphasizing its implications for LoRA's performance. We believe these additions provide deeper insights into the advantages of our approach and significantly strengthen the contributions of the paper.
>
> **References:**
>
> [1] Reece Shuttleworth, Jacob Andreas, Antonio Torralba, and Pratyusha Sharma.                                                 “Lora vs full fine-tuning: An illusion of equivalence.” arXiv preprint arXiv:2410.21228, 2024

---

> > ### Comment · Reviewer_Wiq6 · 2024-12-02
> > **Post rebuttal**
> >
> > I thank the authors for their response. I do want to note the ICLR announcement of the discussion period extension states "The intention of this extension is to allow reviewers and authors to address minor issues, and not for the authors to make major changes to the paper."
> >
> > **W1. Additional Insights**
> >
> > I think that the newly added section of "ENHANCED DIVERSITY IN LORA-ENSEMBLE" has interesting results and analyses. I think this should be highlighted and explained more in future versions. Using a similar analysis Shuttleworth et al. (2024) suggest that LoRA and full finetunning methods have different learning dynamics that make LoRA methods have poor performance OOD. How would this finding affect the current method?
> >
> > **W1.2. Combining ensembling and LoRA**
> >
> > "Science often progresses through multiple groups independently exploring similar ideas" I totally agree and there is room for very similar works, highlighting similarities and dissimilarities is helpful and should be encouraged. One obvious difference is the scale of the experiments, in [A] the LoRA ensembling method is applied in the context of LLMs, where the capacity added by LoRA is insignificant and makes the optimization managable. One potential drawback of the current paper is that applies LoRA method in small scale experiments where the added parameters are important.
> >
> > **W2. Small scale experiment.**
> > If presented , the new experiments would be helpful.
> >
> > **W3. Baselines**
> > Thank you for the additional baseline. I still have some concerns. Were the hyperparameters tuned for the higher rank baseline? Where there any regularization techniques (higher weight decay, early stopping, different learning rate schedule etc.) used for training the higher rank baseline?
> >
> > Overall I tend to maintain my score as the new section deserves more discussions and I still have concerns regarding the scale.

---

> > ### Public Comment · ~Xi_Wang4 · 2024-12-04
> > **Comments from the author LoRA Ensembles**
> >
> > I would like to thank for reviewer and the author for discussing my LoRA ensemble paper.
> >
> > I have a couple of comments regarding the author's comments on my preprint
> >
> > > their implementation details are unclear, and no code is available,
> >
> > This is very true, unfortunately, my code cannot be made public due to legal reasons, I really like the authors' well-organized codebase and I think it can indeed benefit lots of future researchers. Although I may need to point out that the main audience interested in uncertainty + PEFT may be LLM practitioners, it would be nice if the author could additionally include LLM-related experiments.
> >
> > > We also evaluate our method on out-of-distribution scenarios, demonstrating its robustness and generalization
> >
> > Well, my paper also evaluated OOD scenarios, including a couple of uncertainty-related experiments .
> >
> >
> > > Wang et al. work is a concurrent preprint
> >
> > I would like to clarify that my paper is on Arxiv at **29 Sep 2023**, and I will leave the authors and the reviewers to determine the definition of "concurrent work".
> >
> >
> > Nevertheless, my main comments about the paper are:
> >
> > 1. If we look at table 1, the full model ensemble take 16 × 87M ~ 1.5B parameter < 10GB on disk, at the age of LLM, this actually does not sound very significant, and with the 80GB A100s the authors have access to, I can imagine the feasibility of loading all ensemble components on GPU memory concurrently. However, in case the resource is constrained, then the proposed method could be of great usefulness.
> >
> > 2. One of the baseline methods the authors should consider is BatchEnsemble and "rank1 bnn", which consider almost identical settings as this submission, i.e. parameter efficient ensemble for training from scratch. Notice that BatchEnsemble is not very applicable to fine-tuning settings though, since the multiplicative component could ruin the pre-trained weights.

---

### Comment · Area_Chair_rZSf · 2024-11-26
**Encouragement to Actively Participate in the Discussion Phase**

Dear Reviewers,

Thank you for your valuable contributions to the review process so far. As we enter the discussion phase, I encourage you to actively engage with the authors and your fellow reviewers. This is a critical opportunity to clarify any open questions, address potential misunderstandings, and ensure that all perspectives are thoroughly considered.

Your thoughtful input during this stage is greatly appreciated and is essential for maintaining the rigor and fairness of the review process.

Thank you for your efforts and dedication.

---

### Meta-Review · Area_Chair_rZSf · 2024-12-19

**Metareview:**

(a) Summary of Scientific Claims and Findings
This paper introduces LoRA-Ensemble, a parameter-efficient deep ensemble method for self-attention networks. The method leverages Low-Rank Adaptation (LoRA) modules, typically used for fine-tuning, to implicitly create ensemble members by training low-rank matrices while sharing the backbone model’s weights. The method claims to:
Reduce computational and memory costs compared to explicit ensembles.
Achieve superior calibration and accuracy across various classification tasks (e.g., CIFAR-100, HAM10000, ESC-50) and out-of-distribution (OOD) detection.
Exhibit enhanced diversity in ensemble members, contributing to improved epistemic uncertainty modeling.

(b) Strengths of the Paper
Efficiency: The proposed method significantly reduces the computational and memory overhead typically associated with explicit ensembles.
Integration: The use of LoRA modules to create implicit ensembles within self-attention networks is innovative and aligned with trends in efficient fine-tuning.
Broad Applicability within Transformers: Experiments demonstrate the method’s compatibility across multiple Transformer variants, such as ViT and DeiT, and application domains like medical imaging and sound classification.
Improved Calibration: The paper highlights LoRA-Ensemble’s strong performance in epistemic uncertainty modeling and calibration, with empirical results showing improvements over explicit ensembles.

(c) Weaknesses of the Paper
Limited Novelty: The proposed approach closely resembles existing methods, such as BatchEnsemble, which also utilize low-rank adaptations for ensembling. The distinction between LoRA-Ensemble and BatchEnsemble is not convincingly demonstrated, raising concerns about originality.
Claims of novelty are further weakened by a lack of comparisons with key baselines like BatchEnsemble and Masksembles.
Theoretical Gaps: Theoretical justification for why LoRA-Ensemble maintains diversity among ensemble members or improves calibration and uncertainty estimation is lacking.
The absence of analytical insights into the method’s relationship with probability distributions limits its scientific contribution.
Incomplete Evaluation: Experiments are primarily conducted on relatively small-scale datasets, such as CIFAR-100 and HAM10000, which fail to demonstrate the method’s scalability and generalizability to larger, more complex datasets (e.g., ImageNet).
Missing evaluations on text-based tasks, where LoRA is highly relevant, reduce the scope and impact of the work.
Baseline Weaknesses: Critical baselines, such as BatchEnsemble, are omitted, despite their relevance to parameter-efficient ensembling.
The absence of consistent improvements across all benchmarks raises questions about the robustness of the proposed method.

(d) Reasons for Rejection
Limited Novelty and Scope: The similarities with BatchEnsemble, combined with the lack of comparative analysis, significantly undermine the perceived originality of the work. Furthermore, the focus on self-attention networks limits the method’s general applicability.
Weak Theoretical Contribution: The absence of theoretical grounding for the observed performance gains limits the paper’s scientific merit.
Inadequate Evaluation: The reliance on small-scale datasets and the exclusion of key benchmarks (e.g., BatchEnsemble) prevent a comprehensive assessment of the method’s efficacy and scalability.
Unresolved Reviewer Concerns: Despite an extensive rebuttal, key issues—including scalability, baseline omissions, and novelty—remain inadequately addressed.
While LoRA-Ensemble demonstrates potential as a computationally efficient ensembling technique, it lacks the necessary breadth, rigor, and originality to warrant acceptance at ICLR.

**Additional Comments On Reviewer Discussion:**

1. Novelty and Comparison to Existing Methods
Concern: Reviewers noted the method’s similarity to BatchEnsemble and other low-rank adaptation-based techniques, questioning the novelty of LoRA-Ensemble. They highlighted the lack of direct comparisons with these baselines as a significant weakness.
Author Response: The authors argued that LoRA-Ensemble differs in formulation and implementation, particularly in its adaptation to Transformers. They acknowledged the relevance of BatchEnsemble but cited the challenges of adapting it to Transformers as a reason for its exclusion.
Evaluation: This response was not entirely satisfactory. Reviewers maintained that the distinctions between LoRA-Ensemble and BatchEnsemble were insufficiently clear and that the lack of direct comparisons weakened the paper’s claims of originality.

2. Theoretical Analysis
Concern: Multiple reviewers requested theoretical justification for the method’s ability to maintain diversity among ensemble members and improve uncertainty estimation.
Author Response: The authors admitted the difficulty of providing a theoretical analysis and instead added an empirical analysis of diversity metrics, such as pairwise disagreement rates and weight-space cosine similarity.
Evaluation: While the additional analysis was appreciated, the lack of theoretical grounding was viewed as a significant limitation, as it left the method’s performance gains inadequately explained.

3. Evaluation on Larger Datasets and Benchmarks
Concern: Reviewers criticized the evaluation’s reliance on small-scale datasets, such as CIFAR-100 and HAM10000, which do not demonstrate the scalability of the method. They also pointed out missing benchmarks, such as ImageNet and text-based tasks.
Author Response: The authors conducted additional experiments on OOD tasks with CIFAR-100 as in-distribution data and SVHN as OOD data, which showed promising results. They also committed to evaluating the method on iNaturalist but could not include these results due to time constraints.
Evaluation: While the additional OOD experiments strengthened the empirical evaluation, the lack of large-scale benchmarks like ImageNet or text-based tasks remained a significant drawback.

4. Baseline Comparisons
Concern: Reviewers repeatedly requested comparisons with BatchEnsemble and other parameter-efficient ensembling methods, such as Masksembles and PackedEnsembles.
Author Response: The authors acknowledged the omission but argued that adapting these methods to Transformers would require substantial re-engineering, which was beyond the scope of their work.
Evaluation: This justification was not well received, as reviewers felt that these comparisons were critical to establishing the method’s contribution.
Based on these considerations, the paper was deemed not ready for acceptance.

---

### Decision · Program_Chairs · 2025-01-22

Reject